# Modified viral-genetic mapping reveals local and global connectivity relationships of ventral tegmental area dopamine cells

**Kevin Beier\***

Department of Physiology and Biophysics, Neurobiology and Behavior, Biomedical Engineering, Pharmaceutical Sciences, Center for the Neurobiology of Learning and Memory, University of California, Irvine, Irvine, United States

**Abstract** Dopamine cells in the ventral tegmental area (VTA$^{DA}$) are critical for a variety of motivated behaviors. These cells receive synaptic inputs from over 100 anatomically defined brain regions, which enables control from a distributed set of inputs across the brain. Extensive efforts have been made to map inputs to VTA cells based on neurochemical phenotype and output site. However, all of these studies have the same fundamental limitation that inputs local to the VTA cannot be properly assessed due to non-Cre-dependent uptake of EnvA-pseudotyped virus. Therefore, the quantitative contribution of local inputs to the VTA, including GABAergic, DAergic, and serotonergic, is not known. Here, I used a modified viral-genetic strategy that enables examination of both local and long-range inputs to VTA$^{DA}$ cells in mice. I found that nearly half of the total inputs to VTA$^{DA}$ cells are located locally, revealing a substantial portion of inputs that have been missed by previous analyses. The majority of inhibition to VTA$^{DA}$ cells arises from the substantia nigra pars reticulata, with large contributions from the VTA and the substantia nigra pars compacta. In addition to receiving inputs from VTA$^{GABA}$ neurons, DA neurons are connected with other DA neurons within the VTA as well as the nearby retrorubal field. Lastly, I show that VTA$^{DA}$ neurons receive inputs from distributed serotonergic neurons throughout the midbrain and hindbrain, with the majority arising from the dorsal raphe. My study highlights the importance of using the appropriate combination of viral-genetic reagents to unmask the complexity of connectivity relationships to defined cells in the brain.

**\*For correspondence:**
kbeier@uci.edu

**Competing interest:** The author declares that no competing interests exist.

## Editor's evaluation

By addressing shortcomings in rabies viral-mediated labeling of monosynaptic inputs to ventral tegmental area dopamine neurons, this study provides a previously unattained precision of local inputs to VTA dopamine neurons. Main findings include the preservation of a medial to lateral topography in the projection patterns within VTA microcircuitry, prominence of inhibition of DA neurons from the substantia nigra pars reticulata (SNr), DA-DA transmission, and inputs from raphe serotonin neurons. The precise local VTA connectivity described here is important for identifying how DA neurons compute reward, prediction, and movement-related signals during behavior.

## Introduction

Ventral tegmental area (VTA$^{DA}$) cells mediate a variety of motivated behaviors, including reward and aversion (*Björklund and Dunnett, 2007*; *Bromberg-Martin et al., 2010*; *Cohen et al., 2012*; *Lammel et al., 2012*; *Wise, 2004*). Substantial effort has been made to map the brain regions and cell types that provide input to and receive projections from VTA$^{DA}$ cells, a critical step toward understanding

how VTA[DA] cells effect behavioral consequences in response to a variety of stimuli. The recent advent of one-step rabies virus (RABV) has enabled the mapping of inputs onto defined cell types (*Wickersham et al., 2007*). This strategy was employed nearly a decade ago to map brain-wide inputs to DA cells in the VTA and the adjacent substantia nigra pars compacta (SNc) (*Watabe-Uchida et al., 2012*). However, DA neurons in the VTA and SNc are not homogenous, but rather are heterogenous in their molecular signatures, projection patterns, physiological properties, and behavioral functions (*Kim et al., 2016*; *Lammel et al., 2008*; *Lammel et al., 2011*; *Lammel et al., 2012*; *Lerner et al., 2015*). We therefore designed a method, Tracing the Relationship between Inputs and Outputs (TRIO) to map the input–output relationship of projection-defined VTA[DA] cells (*Beier et al., 2015*; *Lerner et al., 2015*; *Schwarz et al., 2015*). TRIO revealed that different subtypes of VTA[DA] cells received biased inputs, and that global input–output maps could be used to infer the behavioral contribution of particular input sites, such as the cortex (*Beier et al., 2015*).

While these and more recent studies have mapped global inputs to VTA[DA] and SNc[DA] cells (*Beier et al., 2015*; *Faget et al., 2016*; *Lerner et al., 2015*; *Menegas et al., 2015*; *Watabe-Uchida et al., 2012*) each study has the same limitation that only inputs located at a distance from the injection site in the midbrain can be assessed. This is because the Cre-dependent TVA protein that facilitates EnvA-mediated infection is also expressed at low levels in non-Cre-expressing cells near the site of injection (*Miyamichi et al., 2013*). Therefore, RABV virions that are pseudotyped with the EnvA protein can infect both Cre-expressing starter cells and non-Cre-expressing cells nearby that express low levels of TVA through leaky, non-Cre-dependent expression. This leaky gene expression is due to incomplete suppression of transcription/translation of TVA. While this may not be an issue with fluorescent molecules or chemogenetic effectors such as DREADDs (*Botterill et al., 2021*), in cases where only small amounts of a gene product are required to exert function, such as TVA, the problem becomes magnified. Only a single functional unit – presumably three TVA molecules bound to an EnvA trimer (*Alsteens et al., 2017*) – is required to enable infection of EnvA-pseudotyped RABV. Injection of adeno-associated viruses (AAVs) expressing TVA and RABV-G 2 weeks prior to injection of EnvA-pseudotyped RABV can result in thousands of infection events near the injection site, even in Cre− animals that should not express TVA (*Beier et al., 2015*). Notably, these infections do not occur if AAVs are not injected, demonstrating that the off-target labeling is TVA-mediated. Therefore, whether RABV-labeled neurons near the injection site are bona-fide inputs to starter cells or cells that are infected via the viral inoculum in previous VTA[DA] mapping experiments cannot be distinguished.

One solution to this problem is to reduce the efficiency of TVA-mediated infection. A mutant version of TVA with a single-point mutation (Glu⁶⁶ → Thr, or TC[66T]) that exhibits only 10% of the efficiency of the wild-type TVA dramatically reduces local off-target labeling (*Miyamichi et al., 2013*; *Rong et al., 1998*). Using this variant enables analysis of the quantitative contribution of local inputs to defined cell types, such as VTA[DA] neurons. Here, I quantified the inputs to VTA[DA] neurons, comparing long-distance inputs to local inputs. Specifically, I quantified the percent of virally labeled inputs to VTA[DA] cells that were located in 57 different brain regions. I used the TC[66T] variant in place of the wild-type TVA, TC[B], in my tracing studies and found that almost half of the inputs to VTA[DA] cells are located near the VTA; many of these represent inputs that were missed or may have been misquantified in the previous studies. I then performed quantitative analyses to identify the biases that local inputs have onto different sets of VTA[DA] cells. I then examined the sources of local GABAergic input to VTA[DA] cells, explored potential connections between DA cells in the ventral midbrain, and quantified the sources of serotonergic inputs to VTA[DA] cells. My analysis is the first of its kind to detail the quantitative contributions of inhibition and neuromodulatory influence from local cells in the midbrain and hindbrain onto VTA[DA] cells, providing a comprehensive global picture of the main cell populations that influence and control VTA[DA] cells.

## Results

I performed one-step RABV mapping using TC[66T] in place of TC[B] (*Miyamichi et al., 2013*; *Figure 1A*). I first injected a combination of Cre-dependent AAVs encoding the RABV glycoprotein, *RABV-G*, and TC[66T] into the VTA of DAT-Cre mice that express the Cre recombinase in DA neurons. Two weeks later, I injected EnvA-pseudotyped, G-deleted, GFP-expressing RABV into the VTA. I then allowed 5 days for RABV spread to input cells before terminating the experiment.

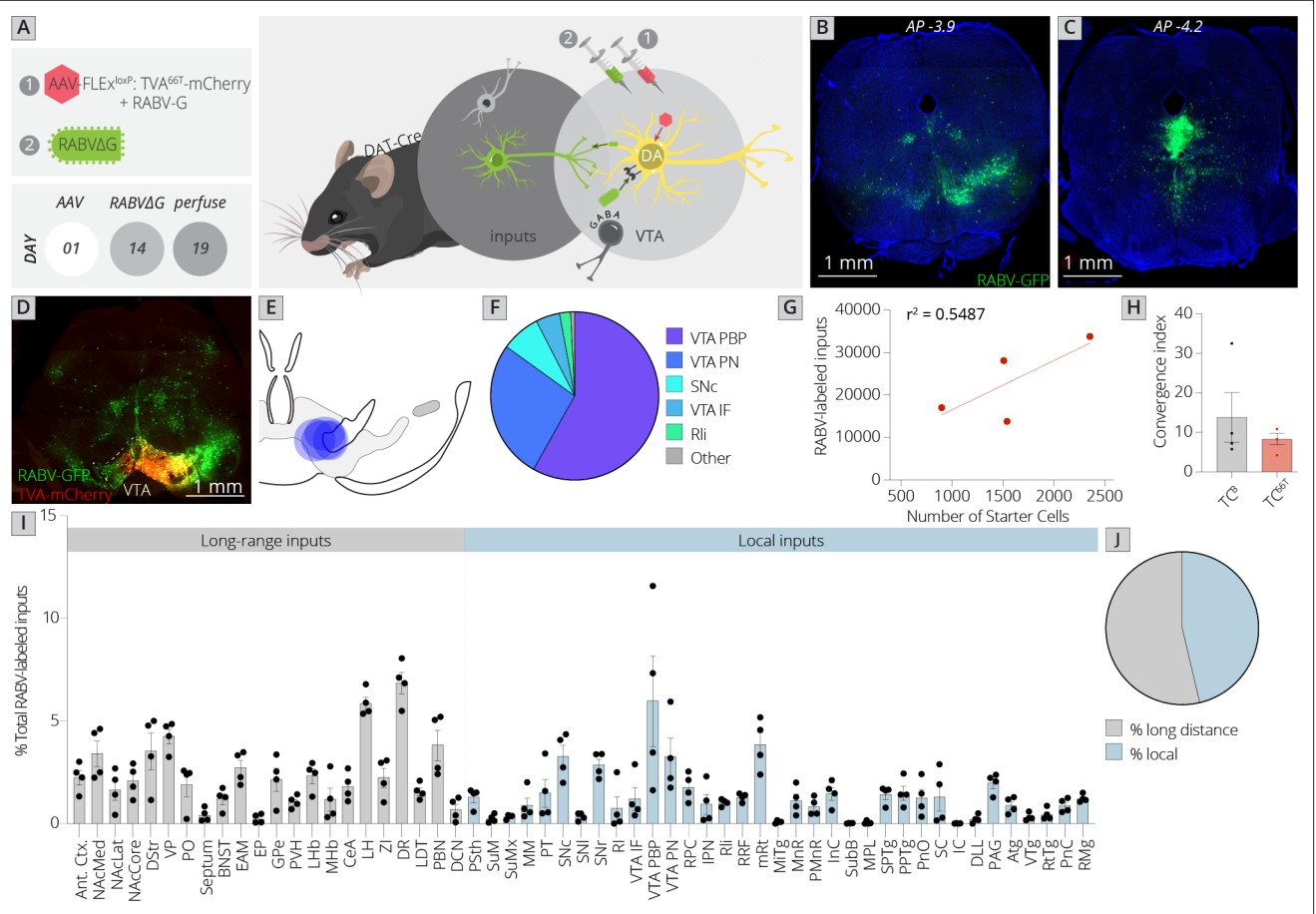

**Figure 1.** Use of a modified viral-genetic strategy to map local and global inputs to VTA[DA] neurons. (**A**) Strategy for viral mapping. On day 1, a combination of adeno-associated viruses (AAVs) expressing the mutated TVA protein fused to mCherry, TC[66T], and RABV-G were injected into the VTA. Two weeks later, EnvA-pseudotyped RABV expressing GFP was injected into the VTA. Animals were sacrificed 5 days later. (**B**) Representative image of local virally labeled inputs at anterior–posterior −3.9 mm from bregma. Scale, 1 mm. (**C**) Representative image of local virally labeled inputs at anterior–posterior −4.2 mm from bregma. Scale, 1 mm. (**D**) Representative image of a midbrain section including starter cells in the VTA as well as local inputs. Scale, 1 mm. (**E**) Starter cell distributions for each of the four experiments. The center of each oval represents the center of mass of starter cells, and the horizontal and vertical radii of the oval represent 1 SD of starter cells in the medial–lateral and dorsal–ventral axes, respectively. (**F**) The fraction of starter cells located in each region in the ventral midbrain. 58% ± 6% were located in the parabrachial pigmented nucleus (PBP), 27% ± 6% in the paranigral nucleus (PN), and 5% ± 2% in the IF nuclei of the VTA, 8% ± 3% in the SNc, 2% ± 1% in the Rli, and 1% ± 0.3% in all other regions. (**G**) Linear regression of RABV-labeled inputs vs. number of starter cells. (**H**) The convergence index, or ratio of inputs to starter cells, for RABV input mapping experiments using TC[B] as reported in *Beier et al., 2015*, or TC[66T]. Only long-range inputs were considered to enable direct comparison between conditions. p = 0.42, 95% CI −21.25 to 10.24, unpaired *t*-test; *n* = 4 for each condition. (**I**) The percentage of each local and long-distance input relative to all RABV-labeled inputs is shown. (**J**) The fraction of inputs that were long-range inputs (54% ± 3%), as mapped by us in the previous studies, and local inputs (46% ± 3%), is shown. All error bars in this figure and others throughout this manuscript represent ±1 standard error of the mean (SEM) (*Beier et al., 2015*).

The online version of this article includes the following figure supplement(s) for figure 1:

**Figure supplement 1.** Quantification of local and long-distance input labeling using TC[B] and TC[66T].

**Figure supplement 2.** UMAP analysis of VTA[DA] input tracing datasets.

Control experiments in non-Cre-expressing mice were performed to examine the local background of non-Cre-mediated, TC[66T]-facilitated infection. We previously observed using TC[B] that, on average, 3,183 cells were infected in a non-Cre-dependent, TC[B]-facilitated fashion per mouse (*Beier et al., 2015*). Using TC[66T], I found an average of 2.67 cells per brain (*Figure 1—figure supplement 1*). This background infection was indistinguishable from controls where no AAV was injected (*Figure 1—figure supplement 1*). When experiments were performed in DAT-Cre mice, between 900 and 2,400

starter cells were observed, and an average of 23,000 total input cells were labeled per experiment (*Figure 1B, C*).

I next examined the location of starter neurons within the ventral midbrain (*Figure 1D–F*). The majority of starter cells were located within the parabrachial pigmented nucleus (PBP: 58% ± 6%) and paranigral nucleus (PN: 27% ± 6%) of the VTA. The rest of the starter neurons were located within the medial aspect of the SNc (8% ± 3%), interfascicular nucleus of the VTA (IF: 5% ± 2%), and rostrolinear nucleus (Rli: 2% ± 1%). These subregions are largely defined by connectivity patterns and cytoarchitectural features of these cells. While not exclusive, DA neurons that project to different forebrain sites are largely located within different substructures of the VTA (*Oades and Halliday, 1987*; *Trutti et al., 2019*). For example, DA neurons projecting to the medial prefrontal cortex and medial shell of the nucleus accumbens are mostly located in the PN, IF, and medial PBP; those projecting to the amygdala are interspersed throughout the PBP; and those projecting to the lateral shell of the nucleus accumbens are largely located in the lateral PBP (*Lammel et al., 2008*). I then examined the ratio of inputs to starter cells, also known as the convergence index, and compared it to tracing performed using TC$^B$. I used only the long-range inputs quantified in my previous studies in order to enable comparison of TC$^{66T}$ with TC$^B$ (*Beier et al., 2015*; *Beier et al., 2019*). The number of RABV-labeled inputs scaled with the number of starter cells, as expected (*Figure 1G*). The average convergence index using TC$^{66T}$ was approximately 8, which was not significantly different from tracing performed using TC$^B$ (*Figure 1H*). Notably, this was also similar to the convergence index of 7 reported by the study from Watabe-Uchida et al. using a nonmutated TVA that reported only a small number of inputs local to the VTA (*Watabe-Uchida et al., 2012*). However, in my study nearly half (46%) of the inputs to VTA$^{DA}$ cells arose from local regions that were not quantified in my previous analysis. Though Faget et al. excluded the VTA, SNc, substantia nigra pars reticulata (SNr), red nucleus (RPC), and interpeduncular nucleus (IPN) likely for precisely the reason that concerned us, these regions combined to yield 20% of the total inputs to the VTA. This included about 10% of the total inputs within the VTA itself, 6% in the adjacent substantia nigra, and 4% in other nearby regions.

However, in addition to this 20% of inputs in regions immediately adjacent to the injection site, given that AAVs can exhibit substantial spread from the injection site I was concerned that non-Cre-dependent, EnvA-mediated infection extended beyond these regions to several other midbrain regions. To assess if this was indeed a problem, I compared my data to that published by Faget et al. for three separate groups of brain sites: (1) those excluded in the previous study, (2) midbrain/hindbrain regions excluded by us previously but not by the previous study, and (3) long-range inputs included by both studies. Using dimensional reduction techniques such as Uniform Manifold Approximation and Projection (UMAP) to compare the overall labeling patterns within these three groups, I found that my data mixed with that from the Faget et al. study for long-range inputs (comparison 3) suggesting that the data were comparable. However, data from the two studies segregated for both the excluded (comparison 1) and questionable (comparison 2) brain regions (*Figure 1—figure supplement 2*). Data mixing in UMAP space supports the conclusion that the variances between the datasets are similar, and thus the datasets can be compared. In contrast, separation of the two datasets in UMAP space suggests that these datasets cannot be readily compared as the variances between input sites in the datasets are different. These results support the likelihood that the method of using a modified, less efficient version of TVA is necessary to properly examine the local input landscape in the ventral midbrain.

## Local inputs to VTA$^{DA}$ cells have differential associations with long-range inputs

Given the large fraction of total inputs to VTA$^{DA}$ that are local, I wanted to explore the nature of these connections and by extension, how they may influence VTA$^{DA}$ cells. Long-range inputs to DA neurons measured using RABV tracing have been published several times (*Beier et al., 2015*; *Faget et al., 2016*; *Watabe-Uchida et al., 2012*). More recently, DA neurons have been subdivided by output site, and the inputs to particular subpopulations compared to one another (*Beier et al., 2015*; *Beier et al., 2019*; *Derdeyn et al., 2021*; *Lammel et al., 2012*; *Menegas et al., 2015*). These analyses have enabled us to understand how input patterns relate to one another, and which inputs are biased onto which sets of VTA$^{DA}$ cells. However, much less is known about the patterns of innervation from local inputs. Previous analyses using dyes and nonviral tracers have been limited by the ambiguity

of not knowing if nearby neurons were labeled from the initial inoculum, or from retrograde tracer spread, and/or if labeled inputs formed synaptic connections with DA cells or rather to other cells or even nearby brain regions. It is generally more difficult to unambiguously label local connections than long-distance ones. This is because previous viral approaches to label monosynaptic inputs have been hampered by off-target AAV expression, while anterograde methods also suffer from a lack of specificity, as it is difficult to exclusively target small regions in the midbrain without contamination from nearby brain regions, including the VTA itself. My approach here was to leverage my previous analyses of long-range inputs to explore with which long-range inputs each set of local inputs associates. This analysis would tell us if local inputs have a medial/lateral bias within the VTA and by extension, if they preferentially target particular sets of DA cells that are located in different subregions of the VTA. If the percentage of inputs from local inputs covaries with long-range inputs that target lateral subregions of the VTA, I would expect the local inputs to also target lateral aspects of the VTA, and if they covary with medial-targeting long-range inputs, I would expect a medial bias in VTA targeting.

I first used UMAP to identify the clustering relationships of both local and long-distance input sites based on all 57 quantified input (22 long distance and 35 local) regions in my RABV tracing data (*Figure 1I*). This approach harnesses intrinsic variation within the dataset, which likely arises from slight differences in injection location between animals. UMAP clusters regions based on those with similar variance across the mice in the dataset; regions that covary across mice likely innervate similar populations of cells and project to similar regions of the VTA. I found that local input sites intermingled with long-distance inputs (*Figure 2A*). This is expected, and suggests that local inputs co-organize with long-distance inputs, rather than being a separate set of circuits. Given that I used only four brains for these UMAP analyses, which is a relatively small number for accurately capturing variance in the dataset, I wanted to test how robust the UMAP embedding was in capturing the relationships between brain regions. I recently performed a similar analysis on a 76-brain dataset that included long-distance RABV-labeled inputs to VTA cells based on projection, neurochemical phenotype, or a combination of these factors (*Derdeyn et al., 2021*). Since UMAP embeddings can be somewhat stochastic due to their reliance on initial seeding conditions, I also computed the distance between points relative to the maximum distance between any two points in each embedding, over 20 embeddings, then averaged across all embeddings for my four brain dataset, as done previously for the 76 brain dataset (*Derdeyn et al., 2021*). I found that the region associations were robust, with only three long-range input regions – the nucleus accumbens medial shell (NAcMed), nucleus accumbens core (NAcCore), and extended amygdala (EAM) associating with different groups of inputs across conditions (*Figure 2B, C*). While the results were not identical, these data suggest that my four brain dataset could be compared with reasonable confidence to my previous dataset. Similar to my 76 brain dataset, I observed three clusters of input regions (*Figure 2D*). The first is the set of brain regions that predominately project lateral or dorsal to the VTA (*Derdeyn et al., 2021*). The regions that project laterally include inputs from the basal ganglia such as the dorsal striatum (DStr), nucleus accumbens lateral shell (NAcLat), globus pallidus external segment (GPe), and cortex. Those that project dorsally include the entopeduncular nucleus (EP), zona incerta (ZI), and deep cerebellar nuclei (DCN). The local inputs that associate with this cluster include mostly those located dorsally and laterally to the VTA, including the periaqueductal gray (PAG), midbrain reticular nucleus (mRT), interstitial nucleus of Cajal (InC), subpeduncular tegmental nucleus (SPTg), subbrachial nucleus (SubB), SNr, and substantia nigra pars lateralis (SNl). To test if the local inputs I identified indeed projected laterally to the VTA, I mapped out the relative innervation by the SNr, mRT, and PAG of the VTA across the medial–lateral gradient. To do this, I used publicly available data from the Allen Mouse Brain Connectivity Atlas analyzed with custom code (*Figure 3A*; *Beier et al., 2019*). In the Allen Mouse Brain Connectivity Atlas, a GFP-expressing AAV is injected into a targeted site, and labeled axons are imaged throughout the brain. By focusing on the medial–lateral axis of the VTA, one can assess if each input site preferentially targets particular areas of the VTA or not. I selected two or three injections for each input brain region. I indeed observed that three selected inputs in cluster 1 – the SNr, mRT, and PAG – displayed a lateral bias in the VTA (*Figure 3B*, *Figure 3—figure supplement 1*).

The second cluster of inputs includes mostly regions that project uniformly across the VTA, including the lateral hypothalamus (LH), paraventricular hypothalamus (PVH), and bed nucleus of the stria terminalis (BNST) (*Derdeyn et al., 2021*). The local inputs associating with this cluster include inputs from all of the subregions of the VTA itself (PBP, PN, and IF), and other inputs located along the midline

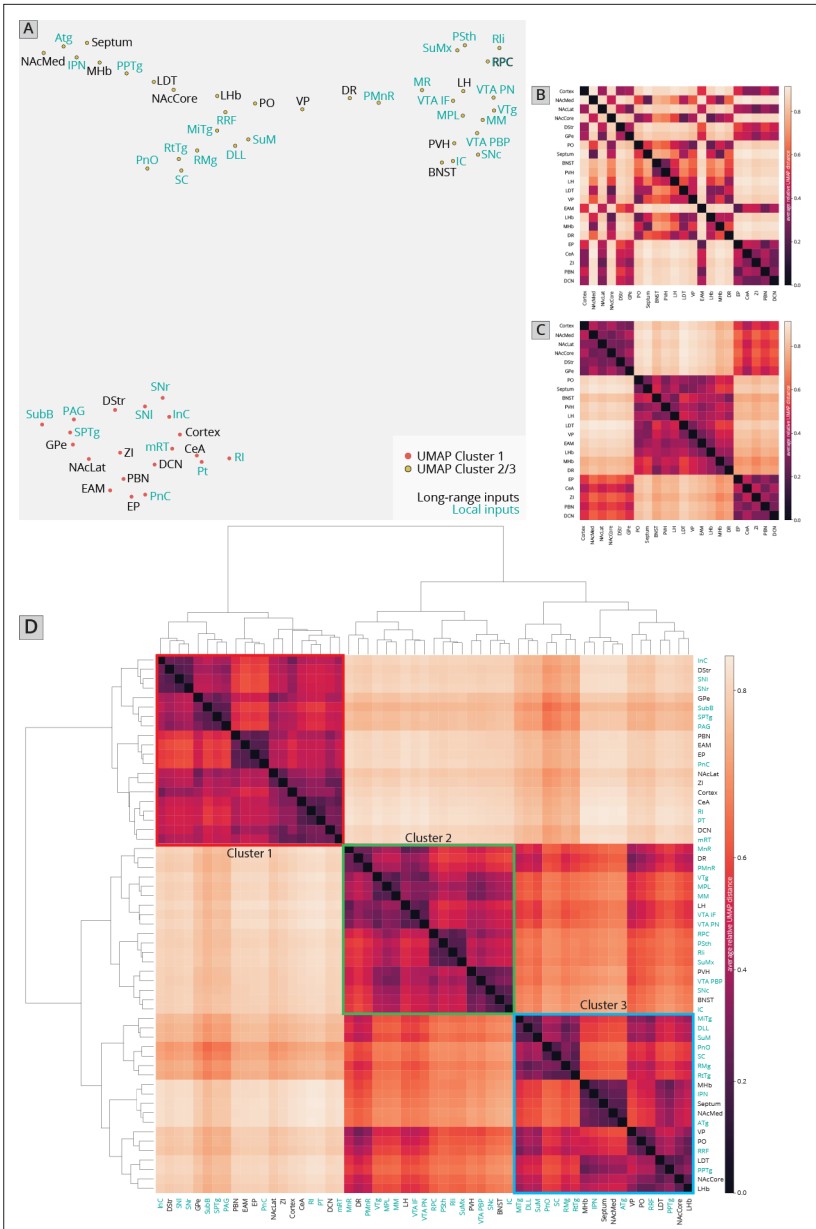

**Figure 2.** Association of local inputs to VTA[DA] cells with long-range inputs. (**A**) Input regions are plotted in Uniform Manifold Approximation and Projection (UMAP) space, embedded with respect to z-scored counts across mouse brains. Clusters represent inputs with similar patterns of variation across the cohort. (**B**) Heatmap of pairwise distances (averaged across 20 UMAP embeddings) for the RABV input data, including only long-distance inputs to enable comparison to our 76-brain dataset, where only long-distance inputs were analyzed. (**C**) Heatmap of pairwise distances from an aggregated UMAP analysis of our 76 brain dataset, from *Derdeyn et al., 2021*, for purposes of comparison. (**D**) Heatmap of pairwise distances for the RABV input data, including both local and long-distance inputs. Regions are grouped according to hierarchical clusters. Clusters are highlighted to match the clusters in the UMAP plot.

[median raphe (MnR), supramammillary decussation (SuMx), medial mammillary nucleus (MM)] as well as those located just laterally to the midline such as the red nucleus (RPC), ventral tegmental nucleus (VTg), inferior colliculus (IC), and parasubthalamic nucleus (PSth). While more varied in their individual projections, the PSth, RPC, and IC show a more consistent projection across the medial–lateral axis of the VTA (*Figure 3C*).

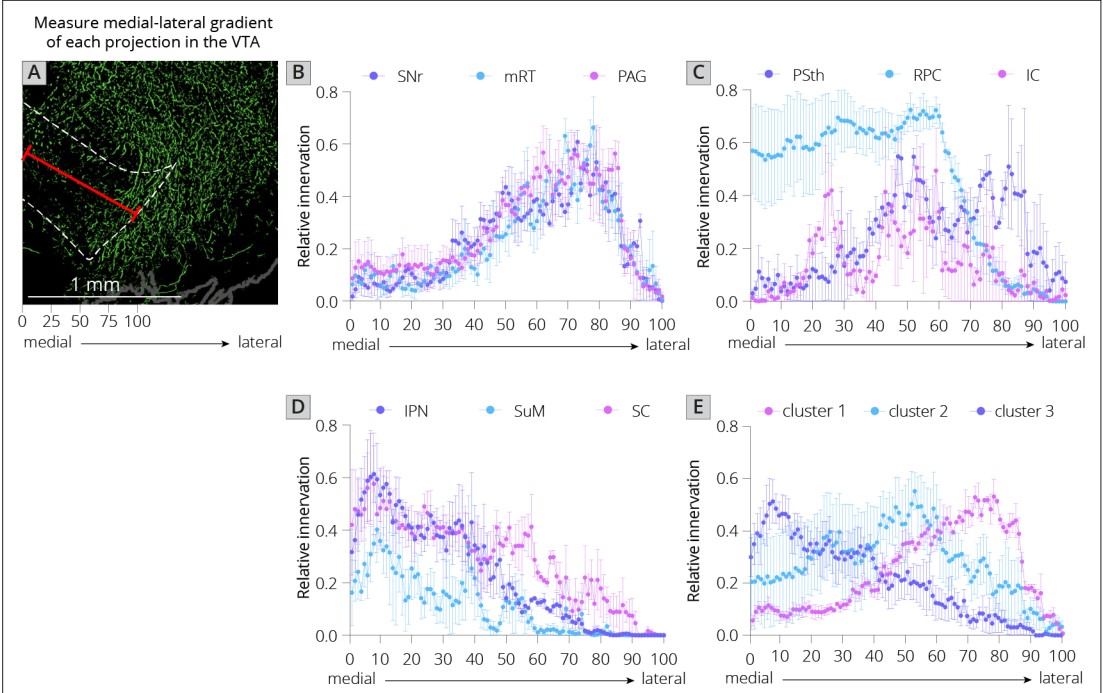

**Figure 3.** Projections from local inputs along the medial–lateral axis of the VTA. (**A**) Schematic of how relative innervation of the medial–lateral axis of the VTA was quantified. Scale, 1 mm. Images used for quantification of the data in panels B–E were obtained from the Allen Mouse Brain Connectivity Atlas. (**B**) Relative innervation across the medial–lateral axis for Uniform Manifold Approximation and Projection (UMAP) cluster 1. (**C**) Relative innervation across the medial–lateral axis for UMAP cluster 2. (**D**) Relative innervation across the medial–lateral axis for UMAP cluster 3. (**E**) Relative innervation across the medial–lateral axis for each UMAP cluster, representing the average of the three individual quantified input regions shown in **B-D**. Error bars represent 1 standard error of the mean (SEM).

The online version of this article includes the following figure supplement(s) for figure 3:

**Figure supplement 1.** Sample images of projections in the VTA from each of the nine quantified local inputs to VTA[DA] cells.

The final cluster of inputs mostly includes those that project medially within the VTA. This includes the medial habenula (MHb) and lateral habenula (LHb), septum, preoptic nucleus (PO), and latero-dorsal tegmentum (LDT). Local inputs associating with this group include midline structures such as the interpeduncular nucleus (IPN), raphe magnus (RMg), supramammillary nucleus (SuM), structures near the midline such as the reticulotegmental nucleus (RtTg), anterior tegmental nucleus (ATg), superior colliculus (SC), slightly more lateral structures such as the retrorubal field (RRF) and pedunculo-pontine tegmental nucleus (PPTg), or inputs with lower counts such as the microcellular tegmental nucleus (MiTg) and the dorsal nucleus of the lateral lemniscus (DLL). The IPN, SuM, and SC all showed a medial bias in the VTA (*Figure 3D*). The difference in medial–lateral preference among the three clusters was especially clear when averaged across the three tested regions and plotted together (*Figure 3E*). These data in sum suggest a topographical organization of inputs to the VTA that applies to both long-distance and local inputs, whereby inputs segregate based on their projection medially or laterally within the VTA, as we have found previously with long-range inputs (*Beier et al., 2015*; *Beier et al., 2019*; *Derdeyn et al., 2021*). Here, I extend these observations by including all inputs to the VTA, regardless of distance from injection site. My approach thus allows us to infer connectivity biases of local inputs that are difficult to ascertain using other viral or nonviral methods.

## Local and distributed GABAergic inhibition to VTA[DA] cells

I next wanted to assess the location of local GABAergic inhibition to the VTA. The majority of studies of inhibition to VTA[DA] cells have focused on either GABA neurons within the VTA or on an anatomically poorly defined structure referred to as the rostromedial tegmental nucleus (RMTg) (*Jhou, 2005*; *Jhou et al., 2009a*; *Jhou et al., 2009b*; *Kaufling et al., 2009*). While extensive efforts have been made to map inputs, including inhibitory cell inputs to the VTA (*Geisler et al., 2007*; *Geisler and Zahm, 2005*;

*Phillipson, 1979*; *Sesack and Grace, 2010*; *Swanson, 2000*; *Zahm et al., 2011*), these methods lacked the connectivity information afforded by RABV. Several studies have performed functional mapping of local inhibition to the VTA. For example, optogenetic stimulation of local GABAergic cells in the VTA inhibits DA cells and is aversive (*Bouarab et al., 2019*; *Tan et al., 2012*; *van Zessen et al., 2012*). These GABAergic populations are likely heterogenous, as they receive differential innervation from input structures and differentially influence different sets of DA cells (*Yang et al., 2018*). Notably, DA neurons located more laterally in the VTA received a stronger spontaneous inhibitory input than DA cells located more medially (*Yang et al., 2018*). Furthermore, the likelihood of a DA cell being inhibited by an aversive stimulus related to the location of the dendrites of those cells, which influenced the number of GABAergic synapses these cells received (*Henny et al., 2012*). Notably, the GABAergic synapse density was higher for dendrites located in the SNr than the SNc, indicating that the location of connections in the ventral midbrain influenced the probability of whether they were excitatory or inhibitory. These data all point to the existence of different subpopulations of local GABA neurons and local GABAergic innervation differentially impacting different populations of DA cells. However, much remains unknown about the sources of GABAergic inputs to DA cells and how they may differentially impact different sets of cells in the VTA. A necessary step to understanding how inhibition impacts subpopulations of DA cells is to identify all of the sources of local inhibition. I

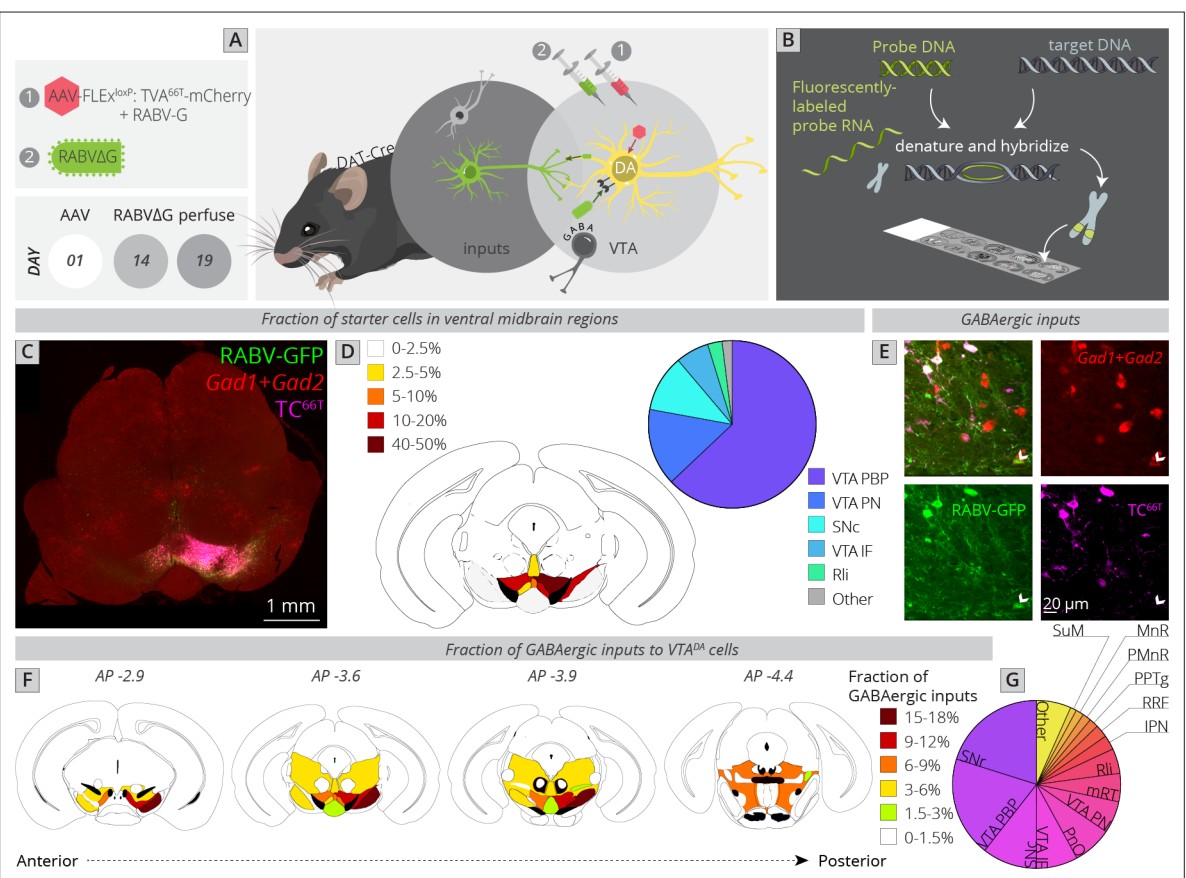

**Figure 4.** Local and distributed GABAergic inhibition to VTA^DA cells. (**A**) Strategy for viral mapping combined with (**B**) fluorescent in situ hybridization with probes for GAD1 and GAD2. (**C**) Sample histology section of the midbrain near the injection site. Green = RABV, red = FISH for GAD1/GAD2, magenta = TC^66T. Scale, 1 mm. (**D**) Breakdown of starter cell location in different midbrain nuclei. 63% ± 1% were located within the parabrachial pigmented nucleus (PBP) of the VTA (15% ± 1% in the VTA PN, 11% ± 2% in the SNc, 6% ± 1% in the VTA IF, 3% ± 1% in the Rli, and 2% ± 0.3% in all other regions). (**E**) Example histology image of labeled cells, with the arrowhead pointing to a RABV+, GAD1/GAD2+, TC^66T− cell. Scale, 50 μM (**F**) Schematic heatmap of GABAergic input location throughout the ventral midbrain. Colors correspond to the total % of local GABAergic inputs located in that particular region. Data from each hemisphere are reported separately. The injection occurred in the right hemisphere. (**G**) Pie chart representing the total fraction of GABAergic inputs located in each region. Data from both hemispheres are combined. 20.1% ± 1.4% of all inhibitory inputs are from the SNr, 19.3% ± 0.7% from the VTA PBP, SNc 10.5% ± 2.4%, VTA IF 8.0 ± 2.6, VTA PN 6.2 ± 0.6, PnO 7.5% ± 2.4%, mRT 5.5% ± 1.3%, Rli 4.8 ± 0.5, IPN 2.8 ± 1.8, RRF 2.5% ± 1.1%, PPTg 1.8% ± 0.8%, PMnR 1.6% ± 0.3%, MnR 1.2% ± 1.2%, SuM 1.2 ± 0.2, and 6.6 ± 0.8 in all other regions.

therefore performed local viral tracing experiments in combination with fluorescent in situ hybridization against the GABAergic markers *Gad1* and *Gad2* in order to provide a quantitative map of inhibition onto VTA$^{DA}$ cells (*Figure 4A, B*).

To assess how this local input map would relate to whole-brain global maps elucidated in *Figure 1*, I first compared the location of starter neurons in the VTA between experiments (*Figure 4C, D*). The majority of starter neurons were located within the PBP nucleus of the VTA (63% ± 1%), followed by VTA PN (15% ± 1%), SNc (11% ± 2%), VTA IF (6% ± 1%), and Rli (3% ± 1%). This was a similar distribution to the whole-brain mapping experiments (*Figure 1F*), enabling direct comparison of these datasets.

I counted all RABV-labeled neurons in regions local to the VTA, and assessed their costaining with GAD1/GAD2 markers as well as TC$^{66T}$. GABAergic input neurons were identified as those that were negative for TC$^{66T}$ and positive for GAD1/GAD2 (*Figure 4E*). I found that, surprisingly, the SNr is the single largest source of inhibition onto the cells that I targeted, representing 20.1% ± 1.4% of all inhibitory inputs (*Figure 4F, G*). Given that the VTA contained 84% of starter cells and SNc contained only 11% of starter cells, and most of these SNc starter cells were on the border of the lateral VTA/medial SNc, these SNr neurons almost certainly inhibit both SNc and VTA neurons. The majority of labeled SNr cells were located in the ventral–medial aspect of the SNr, near to the VTA. The second largest source of input was the PBP nucleus of the VTA (19.3% ± 0.7%), followed by the SNc (10.5% ± 2.4%). These three inputs represented 50% of the total local inhibition of VTA$^{DA}$ neurons. With the addition of the IF (8.0 ± 2.6) and PN (6.2 ± 0.6) nuclei of the VTA, 64% of inhibitory neurons are from regions within or immediately adjacent to the VTA. In addition to inhibitory input from the above regions, VTA$^{DA}$ neurons received inhibitory inputs from distributed sites across numerous regions posterior to the VTA. These included the pontine reticular nucleus, oral part (PnO: 7.5% ± 2.4%), mRT (5.5% ± 1.3%), Rli (4.8 ± 0.5), IPN (2.8 ± 1.8), RRF (2.5% ± 1.1%), PPTg (1.8% ± 0.8%), paramedian raphe nucleus (PMnR: 1.6% ± 0.3%), MnR (1.2% ± 1.2%), and SuM (1.2 ± 0.2). Several of these regions are

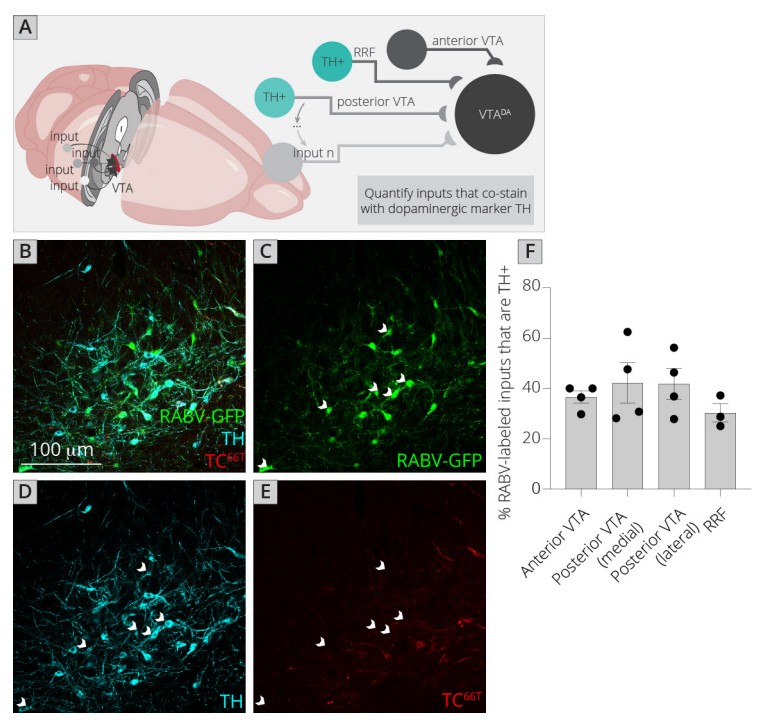

**Figure 5.** Connections between DA cells in the ventral midbrain. (**A**) Schematic of experiments for assessing potential DA–DA connections. Brain sections were stained for TH and mCherry and local inputs in the A8 (RRF) and A10 (VTA) regions were assessed for colabeling with TH and mCherry. Cells that were GFP+, TH+, and mCherry− were considered DAergic inputs. (**B–E**) Example histology image of labeled cells in the RRF. Green = GFP, cyan = TH, red = mCherry. Scale, 100 μm. White arrows indicate RABV+/TH+/mCherry− cells. (**F**) Quantification of TH+ inputs in four different midbrain regions.

thought to contribute to the RMTg, consistent with their known behavioral role in modulating VTA[DA] cell activity. However, it is notable that at least quantitatively, they represent a minor fraction of total inhibitory inputs to VTA[DA] cells.

## Midbrain DA neurons exhibit extensive interconnectivity

Local inputs from the VTA comprised about 10% of the total inputs to VTA[DA] cells (inputs from the VTA PN, VTA PBP, and VTA IF, *Figure 1*), and about 33% of the total local GABAergic input to the VTA[DA] cells (*Figure 4*). However, in addition to the VTA[GABA] input, there is evidence that VTA[DA] cells also receive inputs from local glutamatergic (*Dobi et al., 2010*) as well as DA cells. Physiological evidence has shown that DA neurons release DA within the midbrain via somatodendritic release (*Bayer and Pickel, 1990*; *Groves and Linder, 1983*). Locally released DA then binds D2 autoreceptors expressed on DA cells, suppressing their activity. I wanted to test if DA neurons were connected by conventional means, and if so, what the topology of DA–DA neuron connectivity is in the ventral midbrain.

The challenge of assessing DA–DA connectivity is to distinguish cells that express TVA and thus could serve as starter neurons, and those that did not (and thus represent bona-fide inputs). We previously published the extent of spread of both AAV5-CAG-FLEx[loxP]-TC[B] and AAV8-CAG-FLEx[loxP]-RABV-G in the ventral midbrain given the exact injection coordinates used in this study, which provided a quantified radius of spread for each virus (*Beier et al., 2015*; *Beier et al., 2019*). I stained brain sections with both an anti-tyrosine hydroxylase (TH) antibody to label DA cells, as well as an anti-mCherry antibody to delineate TC[66T]-expressing neurons (*Figure 5A–E*). I then assessed regions outside of the sphere of TVA spread that were either anterior in the VTA, posterior/medial or posterior/lateral in the VTA, as well as in the nearby RRF, and tested for how many neurons costained with TH. I found that about 40% of inputs in each region that did not express clear mCherry protein, even after antibody amplification, costained with TH (*Figure 5F*). That this number did not substantially differ regardless of whether the site was nearer to the injection site (anterior, posterior/medial) or further away (posterior/lateral, RRF) provides further evidence that RABV labeling was not due to direct TVA-mediated uptake, as in that case I would expect a higher percentage of TH+ cells in regions near the injection site. My results suggest that in addition to a substantial GABAergic input from the VTA to local VTA[DA] cells, VTA[DA] neurons receive a large input from other DA cells that comprises approximately 4–5%

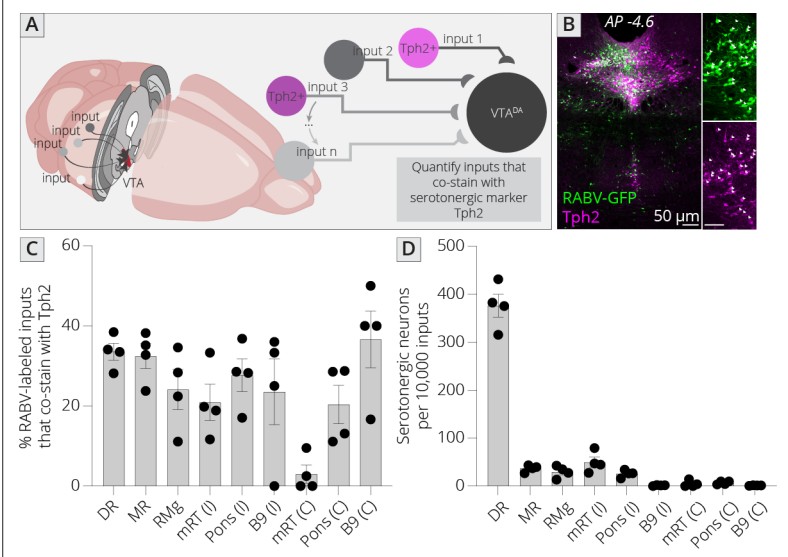

**Figure 6.** Identifying location of serotonergic inputs to VTA[DA] cells. (**A**) Schematic of experiments for assessing potential serotonergic inputs. Brain sections were stained for Tph2, and inputs in the B3, B5, B6, B8, and B9 regions were assessed for colabeling with Tph2. Cells that were GFP+ and Tph2+ were considered serotonergic inputs. (**B**) Example histology image of labeled cells in the midbrain. Green = GFP, magenta = Tph2. Scale, 50 μm. (**C**) Percentage of RABV-labeled inputs in each region that costained with Tph2. (**D**) The number of serotonergic inputs in each quantified region, normalized to 10,000 RABV-labeled inputs. I = ipsilateral, C = contralateral to injection side.

of the total inputs to VTA[DA] cells (11% of total inputs from the VTA/RRF × 40% that are TH+). These results suggest that there is indeed a substantial level of connectivity between DA neurons throughout the VTA and adjacent structures.

## Sources of serotonergic inputs to VTA[DA] cells

The brain's neuromodulatory systems, including cells that release the monoamines DA, serotonin, or norepinephrine, are extensively interconnected (*Hensler et al., 2013*). In particular, serotonin (5-HT) plays a key role in the modulation of the DA system. A number of studies using classical methods such as small molecule and dye tracers have found that the VTA receives strong innervation from cells in the B7 region, also known as the dorsal raphe (DR) (*Phillipson, 1979*). However, the methods employed by these studies do not distinguish connected cells from passing fibers, or inputs connected to non-DA cells. Several studies including my own have demonstrated that serotonergic neurons in the DR do directly connect to both DA and GABA cells in the VTA (*Beier et al., 2015*; *Liu et al., 2014*; *McDevitt et al., 2014*; *Qi et al., 2014*; *Wang et al., 2019*). However, serotonergic input likely arises from other serotonergic nuclei besides the DR. Where the serotonergic neuron input is arising from, what the density of this innervation is, and the fraction of serotonergic innervation provided by each region is not known. Since several of the serotonergic cell-containing brain regions, including the pontine tegmentum (B9) and MnR (B8) regions are located nearby the VTA[DA] cells, connectivity could not be discerned using viral mapping studies that employ the standard FLEx[loxP]/DIO TVA to mark starter cells. Therefore, my approach is uniquely suited to address this question.

I mapped inputs to VTA[DA] neurons, as before, and costained neurons in the midbrain and hindbrain with an antibody to tryptophan hydroxylase 2 (Tph2), which marks serotonergic neurons (*Figure 6A, B*). I examined both hemispheres independently, except for the RMg (B3), DR (B7), and MnR (B8), as these cell populations are located along the midline. I observed that approximately 30% of cells in each region costained with Tph2, with a high of 37% and a low of 20% (*Figure 6C*). This number did not vary substantially, regardless of the region or hemisphere, suggesting that the fraction of cells providing direct connections onto VTA[DA] neurons that are serotonergic within each region is relatively constant across all serotonergic nuclei. I also wanted to assess the fraction of total input provided by each set of serotonergic cells. I therefore normalized the number of serotonergic inputs in each region per 10,000 RABV-labeled input cells. This analysis showed that the vast majority of serotonergic inputs to the VTA indeed arise from the DR (376 ± 24 per 10,000 inputs; *Figure 6D*). The regions providing the next largest quantitative serotonergic inputs are the mRT (B6; 50 ± 11 inputs per 10,000), MnR (37 ± 4 inputs per 10,000), RMg (30 ± 6 inputs per 10,000), and pons (B5; 26 ± 4 inputs per 10,000). Therefore, while the majority of serotonergic inputs do arise from the DR, VTA[DA] cells also receive distributed serotonergic input from several other midbrain and hindbrain nuclei that together represent approximately 1.5% of total inputs to VTA[DA] cells.

## Discussion

In this study, I provide a comprehensive local and global input map to VTA[DA] neurons. I focused on local inputs to the VTA that were inaccessible using previous approaches, and quantified where GABAergic, DAergic, and serotonergic inputs arose in the midbrain and hindbrain. This study provides the first comprehensive map of these connections to VTA[DA] neurons and thus provides a valuable resource for future studies of the VTA as well as a template for mapping both local and global inputs to other structures in the brain.

Inputs to DA cells in the VTA have been previously mapped (*Beier et al., 2015*; *Faget et al., 2016*; *Watabe-Uchida et al., 2012*). However, each of these studies reported a limited number of local input regions, as the issue of local background was likely recognized in each study. The report from Watabe-Uchida et al. included a much reduced local input labeling relative to this study, perhaps due to the limited reporting of input regions. Most importantly, nearly every study to date, including the three cited above, used a wild-type version of the TVA protein that would likely yield hundreds to thousands of non-Cre-dependent, TVA-facilitated infections in the midbrain near the injection site. My approach using TC[66T] to reduce non-Cre-mediated infections near the site of injection enabled high-resolution, low-noise analysis of connectivity to VTA[DA] cells. The fact that nearly 50% of inputs are from local brain sites highlights the importance of local connectivity within the midbrain. That many of the brain

regions are much less studied than their forebrain counterparts represents an opportunity to further study inputs from the midbrain and hindbrain that provide quantitatively substantial inputs to VTA$^{DA}$ neurons yet whose functions remain largely unknown.

## Relationships with different long-range input sites

I previously mapped inputs to VTA neurons based on output site and neurochemical phenotype, amassing a 76-brain dataset that enabled us to dissect the sources that influence patterns of inputs to different cells in the VTA (*Beier et al., 2015*; *Beier et al., 2019*; *Derdeyn et al., 2021*). In this study, I wanted to map the local inputs onto these scaffolds to observe if local inputs have biases toward specific VTA$^{DA}$ subpopulations. I demonstrated here using a subset of local input regions that the three clusters of input regions arising from the UMAP analysis were related to spatial projection to the VTA. I used output tracing experiments from three representative local input regions from the Allen Mouse Brain Connectivity Atlas, as the number I could select was limited. This was due to the fact that most of the local input brain regions are small, and thus without defined Cre-expressing cell populations to target each population, more often than not the desired brain region represented only a small fraction of the total cells expressing GFP. Even so, the brain regions that I identified that did contain at least two injections relatively specifically targeting the desired regions reinforced my interpretation that the UMAP clusters represent different medial–lateral projection patterns in the VTA.

I had previously linked biases in long-range inputs to different VTA cell populations, and showed that these biases were related to stereotyped projection archetypes in the VTA (*Derdeyn et al., 2021*). Namely, inputs that project laterally to the VTA predominantly innervate VTA$^{DA}$ → NAcLat cells, those that uniformly innervate the VTA are biased onto VTA$^{DA}$ → Amygdala cells, those projecting ventromedially preferentially target VTA$^{DA}$ → NAcMed cells, and those that project ventromedially target VTA$^{DA}$ → medial prefrontal cortex (mPFC) cells (*Derdeyn et al., 2021*). Though the four brains used for this analysis were many fewer than the 76 brain dataset used previously, I was able to recapitulate the main associations between brain regions, supporting the validity of my interpretations. This thus suggests that local inputs to cluster 1 (SNr, SNl, mRT, SPTg, SubB, PnC, RI, PT, and InC) principally target VTA$^{DA}$ → NAcLat cells, those in cluster 2 (PSth, IC, RPC, VTA PBP, PN, IF, MnR, Rli, SNc, PMnR, VTg, MPL, MM, and SuMx) predominately innervate VTA$^{DA}$ → Amygdala neurons, and those in cluster 3 (IPN, SuM, SC, PnO, ATg, RMg, PPTg, RRF, RtTg, MiTg, and DLL) are biased onto VTA$^{DA}$ → NAcMed or VTA$^{DA}$ → mPFC cells. Thus, connectivity biases in regions that cannot be easily accessed through traditional retrograde labeling methods as they may be too close to the injection site or highly dependent on precise injection targeting, and also through anterograde labeling methods because the regions are too small to specifically target using for example, AAVs, can be inferred. While many of these connections show a medial–lateral gradient; that is, inputs that are located more medially in the brain project to the medial aspect of the VTA and those located more laterally in the brain project to the lateral VTA, some do not. For example, the PAG is a midline structure, but preferentially innervates the lateral VTA (*Figure 3B*). While there does appear to be associations between functionally related regions, for example the SNl and SNr reside in the same cluster, and VTA IF, PBP, and PN cells together in a different cluster, overall functional relationships between brain regions remain difficult to define. This is partly because many of these midbrain regions are quite small compared to their forebrain counterparts, and their roles are not well defined. This also points to the limitation of combining classic neuroanatomical boundaries with modern neuroscience approaches, a problem that I discuss in further detail later in the discussion.

## DA–DA cell interconnectivity

In addition to local GABAergic inputs, I also found a large number of DA–DA cell connections in the VTA. DA neurons are known to signal to one another through release of DA, which is thought to signal through volume transmission (*Groves and Linder, 1983*; *Hajdu et al., 1973*; *Wilson et al., 1977*). As RABV is thought to transmit through synapses, it is not immediately clear how this DA–DA neuron transmission may occur. In a study mapping connections to direct and indirect pathway neurons in the dorsal striatum, RABV was also observed to transmit to DA cells (*Wall et al., 2013*). However, this mode of transmission occurs approximately 10-fold less efficiently than when RABV was injected directly into the dorsal striatum. Notably, it is not certain that inter-midbrain DA transmission occurs via classic volume transmission and not through focal release/signaling of DA. For example, DA levels

appear to rise to ≥10 µM for brief periods of ≤100 ms (*Ford et al., 2009*). This rapid on/off kinetics of DA signaling is inconsistent with DA signaling at a distance (*Beckstead et al., 2004*). Furthermore, spontaneous exocytotic, GPCR-mediated signaling was observed in SNc$^{DA}$ neurons (*Gantz et al., 2013*). This evidence indicates that, like traditional ligand-gated transmission, GPCR-mediated transmission can occur in a point to point fashion, similar to a classic synaptic mechanism. More recent evidence using the DA sensor dLight and a photoactivatable D2 receptor ligand also supports the hypothesis that DA release occurs from highly specific sites (*Condon et al., 2021*). Although the nature of DA release or contacts between DA cells cannot be inferred from this study, my results also support the possibility of specific contacts between DA cells through which RABV transmission can occur. I estimate that approximately 4–5% of total RABV-labeled inputs are from DA cells. If this number is also influenced by a 10-fold reduction in efficiency of transmission, as estimated from Wall et al., this would indicate that the presence of DA terminals, and thus the potential for DA influence of other VTA$^{DA}$ cells is extraordinarily high.

This interconnectivity of DA cells throughout the ventral midbrain also has implications for tract tracing studies. Several studies, including the earliest input–output viral-genetic mapping study of the VTA defined output cells by injection of RABV into a projection site, and enabled spread of RABV from VTA cells via injection of an AAV expressing RABV-G into the ventral midbrain of DAT-Cre mice (*Lammel et al., 2012*). According to this strategy, any DA cell expressing RABV-G could serve as a starter cell. Thus, for example when RABV was injected into NAcLat, if VTA$^{DA}$ → NAcLat cells receive input from VTA$^{DA}$ → mPFC cells, then RABV could spread to VTA$^{DA}$ → mPFC cells and subsequently, their inputs. This interconnectivity thus has the potential to degrade pathway-specific connectivity. The TRIO strategy was designed to avoid this potential degradation by injection of a third virus, which we chose to be the canine adenovirus (CAV-2) (*Beier et al., 2015*; *Lerner et al., 2015*; *Schwarz et al., 2015*) and later AAV$_{retro}$ or herpes simplex virus (*Ren et al., 2018*). We employed a dual recombinase AND-gate where cell types were defined by Cre expression and outputs defined by expression of Cre-dependent Flp recombinase, delivered via CAV-2. The results from this study suggest that the AND-gate TRIO strategy is required for maintaining pathway specificity in input–output analysis.

## Distributed serotonergic inputs to VTA$^{DA}$ cells

In addition to DA, serotonin plays a key role in the modulation of VTA$^{DA}$ cells. Multiple types of serotonin receptors are expressed in the VTA, including 5-HT1B, 5-HT2A, and 5-HT2C (*Bubar and Cunningham, 2007*; *Doherty and Pickel, 2000*; *Pazos and Palacios, 1985*; *Waeber et al., 1989*). Application of serotonin to the VTA depolarizes DA neurons, likely through 5-HT2A receptors (*Paolucci et al., 2003*; *Pessia et al., 1994*). Studies using classical methods including horseradish peroxidase and radiolabeling found that the VTA receives inputs from brain regions that include serotonergic neurons, including most prominently the DR (*Azmitia and Segal, 1978*; *Hervé et al., 1987*; *Parent et al., 1981*; *Phillipson, 1979*). My results using one-step RABV reinforce these previous findings. However, I also found that DA neurons receive direct innervation from serotonergic neurons located in several other brain sites, including the B3, B5, B6, B8, and B9 regions. The sum of these inputs equates to approximately 1.5% of total inputs to VTA$^{DA}$ cells. While this may seem insubstantial, serotonergic inputs from the DR only represent about 3.7% of total inputs to VTA$^{DA}$ cells, yet these provide a powerful input to VTA$^{DA}$ cells that influences reward behavior by triggering DA release in the NAc (*Liu et al., 2014*; *McDevitt et al., 2014*; *Qi et al., 2014*; *Wang et al., 2019*). It is also possible that this underrepresents the total serotonergic influence on VTA$^{DA}$ cells, as serotonin signaling, like DA, is thought to occur in part by volume transmission, therefore likely reducing the efficiency of RABV labeling. Notably, many serotonergic neurons in the DR also corelease glutamate, which contributes to the rewarding phenotype of stimulation. It is possible that serotonergic neurons in other brain sites may corelease other neurotransmitters as well, which may work in concert with serotonin to modulate the activity and function of VTA$^{DA}$ cells.

## Practical concerns of off-target TVA expression

The use of TC$^{66T}$ enables analysis of inputs near to the site of injection. This approach is required due to high levels of non-Cre-mediated, TVA-facilitated infection of EnvA-pseudotyped RABV. This highlights the issue that investigators regularly analyze brain regions near the injection site without performing the proper controls. The issue is not with the particular AAV vectors used for RABV mapping, but is

representative of problems inherent in all standard AAV vectors. While FLEx$^{loxP}$ and DIO cassettes are typically sufficient to silence expression of fluorescent reporter genes and functional effectors such as DREADDs or optogenetic proteins, low levels of non-Cre-mediated, off-target expression are a problem when only a small amount of gene product is required, such as in the case of recombinase proteins or viral receptors (*Botterill et al., 2021*; *Miyamichi et al., 2013*). In this case, even a low level of expression is sufficient to degrade specificity. Note that in control experiments where Cre-dependent TVA and RABV-G were introduced in non-Cre-expressing mice, we observed many fewer RABV-labeled cells located at a distance from the injection site (*Figure 1—figure supplement 1*). These results suggest that the leak of TVA was sufficient to mediate a substantial amount of infection of RABV locally, as well as some inputs located at a distance from the injection site. Given that we did not observe long-distance inputs labeled when AAV was not injected into the ventral midbrain, it is likely that labeled cells located at a distance from the midbrain were due to retrograde uptake of the TVA-expressing virus, and non-Cre-mediated expression of TVA from these inputs and subsequent RABV infection.

Off-target expression from Cre-dependent vectors has been reported previously. This problem was recently discussed in depth in several recent manuscripts (*Botterill et al., 2021*; *Fischer et al., 2019*). The reasons underlying the off-target expression of transgenes are still unclear, though there are several possibilities. The first is that the DNA construct may experience recombination and revert to the forward orientation. enabling expression of the transgene (*Fischer et al., 2019*). The second is that the viral ITRs may be able to serve as promoters. This would enable gene expression in the noninverted orientation (*Earley et al., 2020*). The ITRs are known to be able to serve as promoters; for example, some gene therapy vectors used the viral ITR as a promoter to express the cystic fibrosis transmembrane conductance regulator gene (*Flotte et al., 1992*). A third possibility is that concate-merization of viral genomes may result in certain genomic configurations that are permissive to tran-scription of the transgene in the correct orientation. For example, if genomes were oriented in the tail–tail configuration, and read-through transcription occurs across the genomes, the transgene may be expressed (*Trapani et al., 2015*; *Yang et al., 1999*). The final possibility is that genomic integration of the AAV genome could occur in regions of the genome where endogenous promoter activity could drive expression of the gene while in the inverse orientation (*Barzel et al., 2015*).

Given that nearly all RABV mapping experiments use wild-type TVA and many analyze virally labeled cells near the injection site, it is important that the issue of nonrecombinase-mediated, TVA-facilitated infection of EnvA-pseudotyped RABV is recognized by the community. As with other tech-niques, application of proper controls should be required for proper interpretation of RABV mapping experiments. I hope that this study serves as an exemplar for the application of such controls and the kinds of questions that can be investigated using the appropriate combination of viral vectors, as I used here to elucidate the nature of inhibitory and neuromodulatory inputs to DA cells in the VTA.

## Alternative strategies for minimizing background labeling

The use of TC$^{66T}$ is one strategy for reducing off-target labeling. There are two other published strat-egies that have been employed to achieve similar results. One is a simple titration of AAV vectors. A recent report indicated that the concentration of helper viruses, including one encoding a Cre-dependent TVA, was critical for obtaining optimal tracing results (*Lavin et al., 2020*). Too high of a concentration of helpers resulted in high levels of off-target labeling in Cre-negative mice, while the efficiency of input labeling dropped off as a function of concentration. Therefore, it is somewhat of a trade-off between off-target labeling and bona-fide input labeling. The second method is to utilize an 'ATG-Out' strategy whereby the initiation ATG codon is moved outside of the lox sites in the FLEx/DIO vector (*Fischer et al., 2019*). This causes the coding region of the gene to be out of frame with the ATG if no Cre-mediated recombination has occurred. The downside of this strategy is that each gene has to be engineered specifically for the particular FLEx/DIO vector into which it is going to be inserted, and the in-frame postrecombination product will have multiple extra amino acids on the N-terminus of the protein.

An additional consideration is that the on- and off-target gene expression via AAV viral vectors can differ substantially depending on several factors. These include the titer of virus that is injected, the promoter that is used, capsid serotype, volume of virus injected, titer, and batch. Reducing viral titer can reduce off-target infection, as shown previously (*Lavin et al., 2020*). Notably, the physical titer of

AAVs is reported in genome copies (gc)/ml. While this is a simple way of titering viruses by detecting the number of genomes present, it does not directly relate to infectious titer, the measure of the number of infectious particles within a preparation. This number is best obtained using several metrics in addition to the standard qPCR (*Grimm et al., 1998*; *Zeltner et al., 2010*). Stronger promoters such as the CAG promoter that I used in this study are generally good for optimizing levels of expression, but higher levels of on-target expression will likely also mean high levels of off-target expression. Different AAV capsid types influence levels of infection and the resulting speed/amount of gene expression. I found that the volume does not substantially increase the spread of virus assuming a relatively slow injection speed, but I found that it did increase the efficiency of viral transsynaptic labeling (*Beier et al., 2015*; *Beier et al., 2019*). Lastly, there can be substantial batch to batch variations in viruses that sometimes exceed other differences. These may be related to the ratio of viral particles: infectious particles, salt concentrations, or other differences between preparations. While these are much harder to quantify, they necessitate that each batch is validated to ensure robust results.

The application of the TC$^{66T}$-mediated approach has several advantages. First, it simplifies the setup and execution of the experiments. Second, though in general a titration of viruses should be performed for each batch to obtain optimal results, a careful titration is not essential for limiting off-target labeling. Third, this strategy does not result in addition of extra amino acids to the N-terminus of the protein, as is necessitated by the ATG-Out strategy. Fourth, I have already compared the convergence index of the TC$^{66T}$-mediated approach and found that it did not significantly differ from tracing initiated using the non-mutated version of TVA (*Figure 1H*). Therefore, large numbers of inputs can be labeled while not suffering from high levels off-target labeling, the trade-off for the viral titration method. Usage of the TC$^{66T}$ variant thus provides several advantages for simplifying experiments while still obtaining optimum tracing efficiency.

## Anatomical definition of VTA subregions and input sites

The RABV-labeled inputs in this manuscript (and most viral mapping studies) are binned into anatomically defined brain regions. The interpretations of input labeling experiments are intrinsically limited by the relative importance of those boundaries. For example, I and others have observed topographical connections from input populations onto VTA cells depending on where the starter cells are in the VTA (*Beier et al., 2015*; *Beier et al., 2019*; *Derdeyn et al., 2021*; *Faget et al., 2016*; *Menegas et al., 2015*; *Watabe-Uchida et al., 2012*). While there is clearly an organization to connectivity within the ventral midbrain, we do not know what factors define these connectivity patterns. For example, I have reported a well-established medial–lateral gradient of inputs to the VTA from the striatum, which is consistent with the ascending spiral hypothesis (*Haber et al., 2000*). However, the molecular underpinnings that form these gradients are not well understood. Presumably, there are different combinations of growth factors and signaling molecules present during development that follow and/or set such gradients and thus influence the ultimate connectivity architecture of the brain, including the VTA. While several sets of axon guidance molecules such as ephrins, netrins, slits, and semaphorins have been identified (*Bashaw and Klein, 2010*; *Yu and Bargmann, 2001*), on a brain-wide scale we know little about how these and other factors govern connectivity patterns. Approaches such as single-cell sequencing combined with viral mapping, for example a recently reported genetically barcoded rabies virus approach (*Saunders et al., 2021*), have great promise for decoding the connectivity logic in the brain. This information will help us to expand our understanding of brain connectivity beyond inputs and outputs in defined brain regions, and would allow connectivity to be viewed as more of a continuum with some constraints based on cytoarchitecture and other features.

In addition to input brain regions being defined by anatomical boundaries, the VTA has also historically been subdivided into several subnuclei (*Oades and Halliday, 1987*; *Trutti et al., 2019*). These subnuclei were divided by differences by connectivity and cytoarchitecture of each region. However, there also is a medial–lateral gradient within the ventral midbrain, including the VTA, that in some ways coincides with and in some way supersedes these divisions. For example, more medial-projecting DA neurons tend to be located more medial in the VTA, while those projecting laterally are located in the lateral VTA. When considering the VTA and its subnuclei, it is important to remember that these anatomical distinctions are classical definitions based on limited information. These boundaries have been useful as a way to contextualize connectivity and functional data; however, given the extensive heterogeneity of VTA cells, they alone are insufficient for understanding how the VTA works.

Nonetheless, these boundaries remain useful as descriptive guideposts that I can use to detail my results, and relate my work to previous studies. Even so, it will be important to reevaluate the utility of these boundaries as more is understood about cell types and connectivity.

### Caveats for interpretation

One important limitation to my study is that I performed analyses on relatively small datasets, typically $n$ = 3 or 4 for each group. This led in some select cases to high levels of variance, for example, in *Figure 1—figure supplement 1* where there is a substantial difference in the local and long-distance non-Cre-dependent labeling. However, in this case, the key point was that the local and long-distance background in the no TVA and TC[66T] controls is near zero, whereas the local and long-distance labeling in the TC[B] injection in the Cre-negative mouse control condition was much higher.

A second limitation in my study is that my spatial analysis was limited to the medial–lateral axis in the VTA. We previously showed that input projections demonstrate both a medial–lateral and dorsal–ventral gradient in the VTA (*Derdeyn et al., 2021*). I chose to focus here on the medial–lateral axis as that is the principal axis of variation for long-range inputs in the VTA (*Derdeyn et al., 2021*). My analyses have generally not had sufficient anterior–posterior resolution since the location of starter cells, which was critical to the original observation of spatial gradients in inputs to the VTA, was obtained using serial 60 µm coronal sections in the VTA (*Beier et al., 2015*; *Beier et al., 2019*). Therefore, the local inputs mapped here may have additional gradients along the dorsal–ventral or anterior–posterior axes that were not captured in my analysis. In addition, the medial–lateral axis does not fully capture the distinction between the VTA IF, PBP, and PN. Therefore, the full variance in connectivity likely varies as a function of spatial location, but does not necessarily have to be continuous across anatomical boundaries.

It is also important to consider the limitations in interpretation of RABV mapping data. One-step RABV mapping or any other retrograde labeling method does not label 100% of inputs to a given cell or cell population. Rather, it has been estimated that RABV labels anywhere between 10% and 40% of inputs from a given cell (*Rossi et al., 2020*; *Wertz et al., 2015*). While this is a wide range, it also indicates that RABV likely labels fewer than half of the actual inputs to a given cell. Over a population, if one assumes that RABV labels inputs in an unbiased fashion, then it would be likely that the method samples most of the inputs that these cells receive. However, we do not know if RABV has input labeling biases – for example for glutamate or GABA neurons, or synapses located proximal or distal to the cell soma. A significant problem is that we do not know the full constellation of inputs for any cell type in the brain. Without this knowledge, it is very difficult to infer any biases that RABV may have, and thus how representative the labeled input population is.

In addition, RABV mapping is often conducted from a population of cells, as done here. In this case, if a given input is labeled from this population of starter cells we do not know which starter neuron it innervates, or if it may innervate more than one starter neuron. Thus, the resolution of the connectivity between inputs and starter cells decreases as the starter cell population increases in number. Lastly, it is not immediately clear how RABV mapping translates to functional properties of connections. For example, we do not know if RABV labels neurons in proportion to particular properties of connections such as synaptic strength or release probability. While we have obtained some evidence that RABV labeling is sensitive to the relative activity of input populations (*Beier et al., 2017*; *Tian et al., 2022*), we only assessed how RABV labeled particular inputs before and after an experience. How input activity more generally may affect RABV labeling is not known. Thus, what can be inferred about functional connectivity from this, or any other RABV mapping study, is limited to the presence of anatomic connections. Nonetheless, these studies are necessary and useful as the basis on which functional connectivity can be assessed to ultimately learn more about how important structures such as the VTA contribute to a variety of normal and pathological behaviors.

## Materials and methods

**Key resources table**

| Reagent type (species) or resource | Designation | Source or reference | Identifiers | Additional information |
|---|---|---|---|---|
| Genetic reagent (*M. musculus*) | Mouse B6.SJL-*Slc6a3*<sup>tm1.1(cre)</sup> (DAT-Cre) | The Jackson Laboratory | RRID:IMSR_JAX:006660 | |
| Antibody | Anti-GFP (chicken polyclonal) | Aves Labs | Cat#: GFP-1020 (RRID:AB_10000240) | IF (1:1000) |
| Antibody | Anti-TH (rabbit polyclonal) | Millipore | Cat#: AB152 (RRID:AB_390204) | IF (1:1000) |
| Antibody | Anti-mCherry (rat monoclonal) | Thermo Fisher | Cat#: M11217 (RRID:AB_2536611) | IF (1:2000) |
| Antibody | Anti-Tph2 (rabbit polyclonal) | Novus Biologicals | Cat#: NB100-74555 (RRID:AB_2202792) | IF (1:1000) |
| Antibody | Anti-chicken AlexaFluor 488 (donkey polyclonal) | Jackson ImmunoResearch | Cat#: 703-545-155 (RRID:AB_2340375) | IF (1:500) |
| Antibody | Anti-rat AlexaFluor 555 (donkey polyclonal) | Jackson ImmunoResearch | Cat#: 712-165-153 (RRID:AB_2340667) | IF (1:500) |
| Antibody | Anti-rabbit AlexaFluor 647 (donkey polyclonal) | Jackson ImmunoResearch | Cat#: 711-605-152 (RRID:AB_2492288) | IF (1:500) |
| Sequence-based reagent | *Gad1* probe | *Weissbourd et al., 2014* | | |
| Sequence-based reagent | *Gad2* probe | *Weissbourd et al., 2014* | | |
| Software, algorithm | Prism 9 | GraphPad, https://www.graphpad.com/scientific-software/prism/ | | |
| Software, algorithm | Fiji | https://imagej.net/software/fiji/?Downloads | | |
| Software, algorithm | Python 3.8 | https://www.python.org/downloads/release/python-380/ | | |
| Other | AAV$_2$-CAG-FLEx$^{loxP}$- TC$^{66T}$ | University of North Carolina, vector core | | Adeno-associated virus, titer: $2.6 \times 10^{12}$ genome copies (gc)/ml |
| Other | AAV$_8$-CAG-FLEx$^{loxP}$-RABV-G | University of North Carolina, vector core | | Adeno-associated virus, titer: $1.3 \times 10^{12}$ gc/ml; |
| Other | RABV$\Delta$G | Made in lab | | Rabies virus, titer: $5.0 \times 10^8$ colony-forming units (cfu)/ml |

## Mice

DAT-Cre mice were obtained from the Jackson Laboratories (*Bäckman et al., 2006*). Mice were housed on a 12-hr light/dark cycle with food and water ad libitum. Six- to eight-week-old mice were used for experiments. Viral vectors were prepared as previously described (*Schwarz et al., 2015*). All procedures followed animal care and biosafety guidelines approved by the University of California, Irvine's Administrative Panel on Laboratory Animal Care and Administrative Panel of Biosafety (AUP-18-163 and AUP-21-125; IBC #2018-1607). Both males and females were used in all experiments in approximately equal proportions.

## Viral tracing

One-step RABV input mapping was performed as previously described, with the substitution of AAV$_2$-CAG-FLEx$^{loxP}$-TC$^{66T}$ in place of AAV$_5$-CAG-FLEx$^{loxP}$-TC$^B$ (*Beier et al., 2015*). Briefly, 100 nl of a 1:1 volume mixture of AAV$_5$-CAG-FLEx$^{loxP}$-TC$^{66T}$ and AAV$_8$-CAG-FLEx$^{loxP}$-RABV-G was injected into the VTA of 6-week-old mice. Two weeks later, 500 nl of EnvA-pseudotyped, GFP-expressing RABV was injected into the VTA. Two weeks for AAV expression is sufficient time to enable robust TC and RABV-G expression and facilitate high-efficiency input tracing (*Beier et al., 2015*; *Beier et al., 2019*;

*Lerner et al., 2015*; *Ren et al., 2018*; *Schwarz et al., 2015*; *Weissbourd et al., 2014*). Five days for RABV spread is sufficient to label thousands of input cells while causing minimal glial labeling, which increases with longer incubation periods. Mice were allowed to recover for 5 days to enable viral spread, after which time experiments were terminated. Data showed describe biological replicates.

The titers of viruses, based on quantitative PCR analysis, were as follows:

$AAV_2$-CAG-FLEx$^{loxP}$-TC$^{66T}$, $2.6 \times 10^{12}$ gc/ml
$AAV_8$-CAG-FLEx$^{loxP}$-RABV-G, $1.3 \times 10^{12}$ gc/ml.

The titer of EnvA-pseudotyped RABV was estimated to be $5.0 \times 10^8$ colony-forming units (cfu)/ml based on serial dilutions of the virus stock followed by infection of the 293-TVA800 cell line. I defined 'local' as within a radius of 1 mm from the injection site in the anterior, medial, and lateral dimensions. My previous quantifications did not include any sites the ventral midbrain posterior to the VTA, even beyond 1 mm, so these were also included in my 'local' tracing analysis, though they were unlikely to have been strongly influenced by background non-Cre-dependent TVA labeling. Experiments were excluded if viral injections deviated substantially from the targeted injection site within the VTA.

## Quantification of convergence index of RABV input mapping

To compute the convergence index, I considered only the long-range inputs to the VTA (those quantified in the previous studies; see *Beier et al., 2015*; *Beier et al., 2019*), as this allowed us to directly compare the efficiency of labeling between TC$^B$ and TC$^{66T}$.

## Immunohistochemistry

Animals were transcardially perfused with phosphate buffered saline (PBS) followed by 4% formaldehyde. Brains were dissected, postfixed in 4% formaldehyde for 12–24 hr, and placed in 30% sucrose for 24–48 hr. They were then embedded in Tissue Freezing media and stored in a −80°C freezer until sectioning. For RABV tracing analysis, consecutive 60 μm coronal sections were collected onto Superfrost Plus slides and stained for NeuroTrace Blue (NTB, Invitrogen). For NTB staining, slides were washed 1 × 5 min in PBS, 2 × 10 min in PBS with 0.3% Triton X-100 (PBST), incubated for 2–3 hr at RT in (1:500) NTB in PBST, washed 1 × 20 min with PBST and 1 × 5 min with PBS. Sections were additionally stained with 4',6-diamidino-2-phenylindole dihydrochloride (DAPI; 1:10,000 of 5 mg/ml, Sigma-Aldrich), which was included in the last PBST wash of NTB staining. Whole slides were then imaged with a ×4 objective using an IX83 Olympus microscope.

For starter cell identification, sections were unmounted after slide scanning, blocked in PBST and 10% NGS for 2–3 hr at room temperature, and incubated in rat anti-mCherry antibody (1:2000, Life Sciences) and rabbit anti-TH antibody (1:1000, Millipore) at 4°C for four nights. After primary antibody staining, sections were washed 3 × 10 min in PBST, and secondary antibodies (donkey anti-rat Alexa 555 and donkey anti-rabbit Alexa 647, Jackson ImmunoResearch) were applied for two nights at 4°C, followed by 3 × 10 min washes in PBST and remounting. Confocal z-stacks were acquired using a ×20 objective on a Zeiss LSM 780 confocal microscope.

## Fluorescent in situ hybridization

We performed fluorescent in situ hybridization experiments as previously described (*Weissbourd et al., 2014*). To make ISH probes, DNA fragments of 400–1000 bp containing the coding or untranslated region sequences were amplified by PCR from mouse whole-brain cDNA (Zyagen) and subcloned into pCR-BluntII-topo vector (Life Technologies, cat# K2800-20). The T3 RNA polymerase recognition site (AATTAACCCTCACTAAAGGG) was added to the 3'-end of the PCR product. Plasmids were then amplified, the insert removed via EcoRI (New England Biolabs, cat #R0101L) digest, and purified using a PCR purification kit (QIAGEN, cat #28104). 500–1000 ng of the DNA fragment was then used for in vitro transcription by using DIG RNA labeling mix (cat #11277073910) and T3 RNA polymerase (cat #11031163001) according to the manufacturer's instructions (Roche Applied Science). After DNase I (Roche Applied Science, cat #04716728001) treatment for 30 min at 37°C, the RNA probe was purified by ProbeQuant G-50 Columns (GE Healthcare, cat# 28-9034-08) according to the manufacturer's instructions. 60 μm consecutive sections were collected onto Superfrost slides (no coating, Fisher Scientific, cat #22-034-980), dried, and stored at −80°C until use. Specific slides were then thawed and viewed on an Olympus compound fluorescence microscope,

and the sections containing regions of interest were recorded. Those sections were then floated off using PBS into wells of a 24-well plate. The sections were fixed for 15 min in 4% formaldehyde in PBS at room temperature, rinsed with PBS, and incubated with 7 µg/ml Proteinase K (Life Technologies, cat #25530-049) in 10 mM Tris–Cl, pH 7.4, 1 mM EDTA for 10 min at 37°C. After fixing again with 4% formaldehyde in PBS for 10 min and rinsing with PBS, the sections were incubated with 0.25% acetic anhydride in 0.1 M triethanolamine, pH 8.0, for 15 min and washed with PBS. Probes were diluted (~1:1000) with the hybridization buffer (50% formamide, 10 mM Tris–Cl pH 8.0, 200 µg/ml tRNA, 10% Dextran Sulfate, 1× Denhalt's solution, 600 mM NaCl, 0.25% SDS), mixed well, preheated at 85°C for 5 min, and applied to each well (300–500 µl/well). After 16–20 hr of incubation at 50°C, the sections were washed, first with 2× SSC-50% formamide, then with 2× SSC, and finally with 0.2× SSC twice for 20 min at 65°C. After blocking for 1–2 hr with the 1% blocking reagent (Roche Applied Science, cat# 10057177103), sections were incubated with alkaline phosphatase-conjugated anti-DIG antibody (1:1000, Roche Applied Science, cat# 1093274) and chicken anti-GFP antibodies (1:500; Aves Labs, cat# GFP-1020) overnight at 4°C. After washing with Roche Wash Buffer (cat# 11585762001) three times for 15 min followed by rinsing with the detection buffer (100 mM Tris–Cl, pH 8.0, 100 mM NaCl, 10 mM $MgCl_2$), probe-positive cells were detected by Fast Red TR/Naphthol AS-MX Tablets (Sigma-Aldrich, cat# F4523). After washing with Roche Wash Buffer three times for 10 min, sections were incubated with FITC-conjugated donkey anti-chicken antibodies (1:200; Jackson ImmunoResearch) for an additional 1–2 hr, and washed with PBS three times for 10 min. Finally, the sections were treated with PBS containing (DAPI Sigma-Aldrich, cat# D8417) for 20 min and mounted with cover glass using Fluorogel (Electron Microscopy Sciences, Cat#17985-10). Sections were imaged by confocal microscopy (Zeiss 780). Images were processed in ImageJ. We used the Cell Counter plugin in FIJI to quantify overlap.

## Dimensional reduction of RABV input data

UMAP was used as a nonlinear dimensional reduction technique on input data. UMAP is optimized for finding local and global structures in high-dimensional data. Analyses were performed using the official UMAP library (*McInnes et al., 2018*). The fractional counts data were z-scored to compare variation in output and input sites across regions with different magnitudes of counts. Z-scored data were dimensionally reduced with UMAP to find clusters of input sites with similar patterns of variation. UMAP parameters were tuned manually to optimize stability of clusters.

## Analysis of fluorescent axonal labeling from Allen Mouse Brain Connectivity Atlas

Analysis was performed as previously described (*Beier et al., 2019*). Data from three separate injections for each local VTA[DA] input site were analyzed. Experiments had to show GFP labeling in the VTA. For each experiment, three images of the VTA, each spaced approximately two sections apart, were captured at screen resolution. In ImageJ, using the line tool at a thickness of 100, a line was drawn from the midline to the end of the medial lemniscus running ventrolateral through the VTA, and the gray value (a metric of axon coverage) was obtained as a function of distance from the midline. To normalize these values, data were run through custom MATLAB code to segment data into 100 bins and normalized to the maximum intensity value for that image. The three normalized values were then averaged for each brain, and the data from three separate brains then were averaged into one condition and plotted as a percentage across the medial–lateral axis.

### Statistical analysis

Unpaired t-tests were used to compare means between groups (*Figure 1H*).

## Acknowledgements

I would like to thank Pieter Derdeyn for assistance with UMAP analysis, and May Hui for assistance with figure production. This work was funded by NIH DP2-AG067666, R00-D041445, R01-DA054374, TRDRP T31KT1437, and T31IP1426, One Mind OM-5596678, Alzheimer's Association AARG-NTF-20-685694, New Vision Research CCAD2020-002, and ADPA APDA-5589562.

## Additional information

### Funding

| Funder | Grant reference number | Author |
| --- | --- | --- |
| National Institutes of Health | DP2-AG067666 | Kevin Beier |
| National Institutes of Health | R00-D041445 | Kevin Beier |
| National Institutes of Health | R01-DA054374 | Kevin Beier |
| Tobacco-Related Disease Research Program | T31KT1437 | Kevin Beier |
| Tobacco-Related Disease Research Program | T31IP1426 | Kevin Beier |
| One Mind | OM-5596678 | Kevin Beier |
| Alzheimer's Association | AARG-NTF-20-685694 | Kevin Beier |
| New Vision Research | CCAD2020-002 | Kevin Beier |
| American Parkinson Disease Association | APDA-5589562 | Kevin Beier |

The funders had no role in study design, data collection, and interpretation, or the decision to submit the work for publication.

### Author contributions

Kevin Beier, Conceptualization, Data curation, Formal analysis, Funding acquisition, Investigation, Methodology, Project administration, Resources, Software, Supervision, Validation, Visualization, Writing – original draft, Writing – review and editing

### Author ORCIDs

Kevin Beier http://orcid.org/0000-0002-4934-1338

### Ethics

This study was performed in strict accordance with the recommendations in the Guide for the Care and Use of Laboratory Animals of the National Institutes of Health. All of the animals were handled according to approved institutional animal care and use committee (IACUC) protocols (AUP-18-163 and AUP-21-125) of the University of California, Irvine. All surgery was performed under isoflurane anesthesia, and every effort was made to minimize suffering.

### Decision letter and Author response

Decision letter https://doi.org/10.7554/eLife.76886.sa1
Author response https://doi.org/10.7554/eLife.76886.sa2

## Additional files

### Supplementary files

• MDAR checklist

### Data availability

All data and materials generated or analyzed during this study are included in the manuscript and supporting files. Data source files are available as Figure 1—source data 1, Figure 4—source data 1, Figure 5—source data 1, and Figure 6—source data 1. Analysis of previously published data is included in Beier et al., 2015 and Beier et al., 2019 (relevant for Figure 2). All custom code is available on GitHub. Raw data are available on Zenodo (https://doi.org/10.5281/zenodo.6407281), as is our custom UMAP code (https://doi.org/10.5281/zenodo.6565006).

The following dataset was generated:

| Author(s) | Year | Dataset title | Dataset URL | Database and Identifier |
|---|---|---|---|---|
| Beier KT | 2022 | Modified viral-genetic mapping reveals local and global connectivity relationships of ventral tegmental area dopamine cells | https://doi.org/10.5281/zenodo.6407281 | Zenodo, 10.5281/zenodo.6407281 |

The following previously published dataset was used:

| Author(s) | Year | Dataset title | Dataset URL | Database and Identifier |
|---|---|---|---|---|
| Schwarz LA, Kremer EJ, Malenka RC, Luo L | 2015 | Inputs to DA cells | https://ars.els-cdn.com/content/image/1-s2.0-S0092867415008521-mmc2.xlsx | content, S0092867415008521-mmc2 |

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
