## [Editor Report]

By addressing shortcomings in rabies viral-mediated labeling of monosynaptic inputs to ventral tegmental area dopamine neurons, this study provides a previously unattained precision of local inputs to VTA dopamine neurons. Main findings include the preservation of a medial to lateral topography in the projection patterns within VTA microcircuitry, prominence of inhibition of DA neurons from the substantial nigra pars reticulata (SNr), DA-DA transmission, and inputs from raphe serotonin neurons. The precise local VTA connectivity described here is important for identifying how DA neurons compute reward, prediction, and movement-related signals during behavior.

---

## [Decision Letter]

**Decision letter after peer review:**

Thank you for submitting your article "Modified viral-genetic mapping reveals local and global connectivity relationships of ventral tegmental area dopamine cells" for consideration by *eLife*. Your article has been reviewed by 3 peer reviewers, one of whom is a member of our Board of Reviewing Editors, and the evaluation has been overseen by Kate Wassum as the Senior Editor. The following individual involved in the review of your submission has agreed to reveal their identity: Joshua Tate Dudman (Reviewer #3).

Essential revisions:

(1) For maximal impact and for adherence to *eLife* reporting standards it will be essential to release as much of these data as possible into the public domain, including at the minimum detailed tables with raw data quantification, but also consider sharing analysis code (e.g., for UMAP), as well as image files.

(2) Please clarify important points raised below. No new experiments are necessary – but please provide thoughtful responses to the following requests and issues.

2.1 Provide more background/references to explain the problem of low background levels of TVA expression. In theory, there should be zero Cre-independent expression of a functional TVA. Can the author please provide some explanation, or clarification, as to why this Cre-independent TVA expression (even if only a miniscule quantity) actually happens and is unavoidable? The paper is really all about the usage of the TC66T TVA variant, so this section should educate readers on the relevant nuances.

Also please include references to other strategies to address this problem. Previously published studies have attempted to address the problem of low background levels of TVA expression and non-Cre dependent labeling of inputs near injection sites. One solution to prevent this ectopic expression of TVA is to titrate concentration of helper virus for efficient labeling of starter cells while minimizing non-Cre mediated expression, e.g. Wickersham's group https://doi.org/10.3389/fnsyn.2020.00006. Overall, please include a discussion of alternative, potentially simpler solutions. Please address how volume, titer, AAV serotype, promoter, and other batch effects could alter relative to AAV helper virus expression change the background level of non-Cre dependent TVA infection. Without varying these parameters extensively, especially given the diverse applications of this technology in hundreds of neuroscience labs, it is challenging to evaluate the depth of the problem with mapping local inputs.

2.2 Please clearly define what is and what is not meant by the word 'input'

Throughout the paper the author reports % of inputs as a quantity. But an 'input' can mean many things and more clarity is required to reduce confusion. Ideally, a report of % of inputs to a VTA DA neuron would relate to the actual fraction of inputs. For example if a given VTA neuron had 100 synapses, what fraction comes from SNR or STN or VTAgaba or VTAda? This is a per-neuron value, but even across a population of neurons an equivalent analysis could hold – if 100 neurons collectively receive 10,000 inputs, what fraction of those synapses come from a given location. This notion of input is not what is being reported here. Instead, if I understand correctly, it's the number of retrogradely labeled neurons from a given 'area' divided by the total number of retrogradely labeled neurons. It's unclear what physiological relevance this value has, for several reasons. First, a given neuron could project to anywhere between 1-100% of the starter cells. In the present analysis it's an input either way. Second, a given single neuronal input could be weak or strong, distal dendrite or somatic. Given the methods of this paper, synaptic strength is irrelevant. The author is likely fully aware of these issues – which may seem obvious – but I think they should be spelled out in the introduction when the word 'input' is defined, and again in the discussion as caveats for interpretation.

Third, the fraction of inputs in this paper is labeled as per anatomical region, established in the literature, but possibly meaningless for function. For example, eyeballing the figures from this paper or even the past work show a continuous smear of VTA-projecting neurons that follow various mediolateral, dorsoventral, anteroposterior gradients. The 'boundaries' between areas may or may not reflect anything meaningful beyond the history of neuroscientists drawing lines. Single cell sequencing and better pathway specific demarcation will reinforce the validity of these lines or reveal them to be irrelevant. Again, the only request for future versions of this paper is that caveats be made about how seriously to accept these boundaries. For example, does a VTA DA projecting neuron in the STN that is part of a continuous smear of neurons that reaches into the LH, 'know' that it is in the STN – especially if the inputs to these structures are also overlapping? Ideally the author could contextualize the findings from this work into some principle of organization of neuronal projections into VTA – beyond a simple well established adherence to a weak topography that respects mediolateral gradients.

2.3 For this paper – this last issue also relates to confusion over the subterritories of VTA, where it is claimed that starter cells were found in "PBP nucleus of the VTA, 15 {plus minus} 1% in the VTA PN, 11 {plus minus} 2% in the SNc, 6 {plus minus} 1% in the VTA IF, 323 3 {plus minus} 1% in the Rli, and 2 {plus minus} 0.3% in all other regions." Yet the figure shows a continuous smear of neurons without clear nuclei or boundaries. While it is laudable that the author is attempting to achieve rigor of identification within VTA sub territories, more citations or background is necessary to justify them. And how do these acronyms relate to the catecholamine A1-A13 nomenclature? Are these boundaries just lines drawn by past researchers? Are they supported by clearcut distinctions in projection patterns, by distinct genetic identity as revealed by single cell sequencing? More clarification and background is necessary for VTA subdivisions, ideally with how they relate (or don't) to distinct targets. Finally, even when this issue of VTA sub-territories is addressed, the delineation of VTA into these subterritories is out of synch with the later portion of the paper that motivates mediolateral distinctions in input/output. Can these sections be better unified?

2.4 In Figure 2, the author's use of dimensionality reduction to analyze the relationship across local and long-range inputs is compelling and offers new ways of dissecting large-scale monosynaptic rabies tracing data. The author used z-score of fractional counts for UMAP input data and this ultimately aligns to spatial location of input fibers within the VTA. Considering that the UMAP input data is fractional counts for the number of input cells, does the corresponding spatial location of fibers relate to the density of fibers or fibers that are contacting specific subsets of VTA neurons?

2.5. The conclusion that local inputs also reflect long range anatomical gradients is exciting and interesting, but it can be better highlighted. The below quote for example packs lots of information into a few clauses with relatively little discussion.

"This thus suggests that local inputs to cluster 1 (SNr, SNl, mRT, SPTg, SubB, PnC, RI, PT, InC) principally target VTADA→NAcLat cells, those in cluster 2 (PSth, IC, RPC, VTA PBP, PN, IF, MnR, Rli, SNc, PMnR, VTg, MPL, MM, SuMx) predominately innervate VTADA→Amygdala neurons, and those in cluster 3 (IPN, SuM, SC, PnO, ATg, RMg, PPTg, RRF, RtTg, MiTg, DLL) are biased onto VTADA→NAcMed or VTADA→mPFC cells. "

Please expand this section, perhaps in the discussion, to include potential functional significance and/or discovery of developmental principles consistent with these results.

Please ensure your manuscript complies with the *eLife* policies for statistical reporting: https://reviewer.elifesciences.org/author-guide/full "Report exact p-values wherever possible alongside the summary statistics and 95% confidence intervals. These should be reported for all key questions and not only when the p-value is less than 0.05."

If you have not already done so, please include a key resource table.

*Reviewer #1 (Recommendations for the authors):*

While none of the results are particularly surprising or dramatically change the way this reviewer thinks about VTA function, rigorous identification of local inputs to VTA DA neurons is important for the field.

Major concerns relate to issues of clarity and presentation of the methodology.

(1) My understanding is that in theory at least, there should be zero Cre-independent expression of a functional TVA. Yet the author both cites and shows that there is enough to enable RABV to be taken up by non-Cre expressing cells. Can the author please provide some explanation, or clarification, as to why this Cre-independent TVA expression (even if only a miniscule quantity) actually happens and is unavoidable? The paper is really all about the usage of the TC66T TVA variant, so this section should educate readers to the relevant nuances.

(2) Throughout the paper the author reports % of inputs as a quantity. But an 'input' can mean many things and more clarity is required to reduce confusion. Ideally, a report of % of inputs to a VTA DA neuron would relate to the actual fraction of inputs. For example if a given VTA neuron had 100 synapses, what fraction comes from SNR or STN or VTAgaba or VTAda? This is a per-neuron value, but even across a population of neurons an equivalent analysis could hold – if 100 neurons collectively receive 10,000 inputs, what fraction of those synapses come from a given location. This notion of input is not what is being reported here. Instead, if I understand correctly, it's the number of retrogradely labeled neurons from a given 'area' divided by the total number of retrogradely labeled neurons. It's unclear what physiological relevance this value has, for several reasons. First, a given neuron could project to anywhere between 1-100% of the starter cells. In the present analysis it's an input either way. Second, a given single neuronal input could be weak or strong, distal dendrite or somatic. Given the methods of this paper, synaptic strength is irrelevant. The author is likely fully aware of these issues – I think they should be clearly spelled out in the introduction when the word 'input' is defined, or perhaps in the discussion as caveats for interpretation.

(2.1) Third, the fraction of inputs in this paper is labeled as per anatomical region, established in the literature, but possibly meaningless for function. For example, eyeballing the figures from this paper or even the past work show a continuous smear of VTA-projecting neurons that follow various mediolateral, dorsoventral, anteroposterior gradients. The 'boundaries' between areas may or may not reflect anything meaningful beyond the history of neuroscientists drawing lines. Single cell sequencing and better pathway specific demarcation will reinforce the validity of these lines or reveal them to be irrelevant. Again, the only request for future versions of this paper is that caveats be made about how seriously to accept these boundaries. For example, does a VTA DA projecting neuron in the STN that is part of a continuous smear of neurons that reaches into the LH, 'know' that it is in the STN – especially if the inputs to these structures are also overlapping? Ideally the author could contextualize the findings from this work into some principle of organization of neuronal projections into VTA – beyond a simple well established adherence to a weak topography that respects mediolateral gradients.

For this paper – this last issue also relates to confusion over the subterritories of VTA, where it is claimed that starter cells were found in "PBP nucleus of the VTA, 15 {plus minus} 1% in the VTA PN, 11 {plus minus} 2% in the SNc, 6 {plus minus} 1% in the VTA IF, 323 3 {plus minus} 1% in the Rli, and 2 {plus minus} 0.3% in all other regions." Yet the figure shows a continuous smear of neurons without clear nuclei or boundaries. While it is great that the author is attempting to achieve rigor of identification within VTA sub territories, more citations or background is necessary to justify them. And how do these acronyms relate to the catecholamine A1-A13 nomenclature? Are these boundaries just lines drawn by past researchers? Are they supported by clearcut distinctions in projection patterns, by distinct genetic identity as revealed by single cell sequencing? More clarification and background is necessary for VTA subdivisions, ideally with how they relate (or don't) to distinct targets.

Finally, even when this issue of VTA sub-territories is addressed, the delineation of VTA into these subterritories is out of synch with the later portion of the paper that motivates mediolateral distinctions in input/output. Can these sections be better unified?

(3) Signal to noise is a reported quantity in comparing the TC66T to the TCB TVA variants. It is not clear what this ratio relates to, what is at stake in the quantity. The methods section does not reduce confusion. Why 1.87? Are 40,955 'inputs' equal to 40,955 distinct cells that were counted? In how many animals? Is a single cell a single input?

*Reviewer #2 (Recommendations for the authors):*

General:

For maximal impact it would be essential to release as much of these data as possible into the public domain, including at the minimum detailed tables with raw data quantification, but also consider sharing analysis code (e.g., for UMAP), as well as image files.

Related to Figure 3:

Would UMAP analyses of just GABAergic fractional counts also reveal clusters of inputs that could also map to the relative innervation of DA neurons?

Other:

– Several figures have the smallest font of illegible size, even with zoomed in review.

– Formalize "local" and "long distance" in microns from the injection site.

– State the volume of rabies virus injected in the method section.

– Many previous studies inject rabies virus three weeks after helper virus, but this study waits only 2 weeks. What is the justification for this?

– The legend for supplemental figure 3 is vague. It lacks scale bars and does not specify what the thick gray lines are, or difference between image on left vs. right.

– Lines 350-351, Figure 1 does not explicitly show that 10% of total inputs to VTA DA neurons are from local GABAergic neurons.

– Figure 3C is missing a scale bar.

– Figure 4b-e, images are missing scale bar and the legend does not specify what the white arrows represent.

– Line 367, "nearly" should be "nearby"?

*Reviewer #3 (Recommendations for the authors):*

Comparisons to prior data with UMAP was tough to draw conclusions from (Supp Figure 2) as noted by the authors. However, the augmented analysis using recent work was nice and indeed built confidence in the conclusion that the datasets were likely comparable. A little clarity about how conclusions were drawn would be helpful. Of course additional subjects to increase the N would be helpful, but understandable that Bender chose not to do further samples.

"To test if the local inputs we identified indeed projected laterally to the VTA, we mapped out the relative innervation by the SNr, mRT, and PAG of the VTA across the medial-lateral gradient, as done previously for long-range inputs, using the Allen Mouse Brain Connectivity Atlas (Figure 2E) (Beier et al., 2019). We indeed observed that three selected inputs in cluster 1 – the SNr, mRT, and PAG – displayed a lateral bias in the VTA (Figure 2F, Supplemental Figure 3). While extensive efforts have mapped the inputs, including inhibitory cell inputs to the VTA (Geisler et al., 2007; Geisler and Zahm, 2005; Phillipson, 1979; Sesack and Grace, 2010; Swanson, 2000; Zahm et al., 2011), these methods lacked the connectivity information afforded by RABV."

I agree with this point and just use this as a jumping off point. There is a reasonable number of studies that have attempted to measure functional connectivity with electrophysiology/optogenetic or reconstruction of axons from local GABAergic neurons (e.g. from SNr) and yet those papers are little discussed or cited. Might be relevant given that they arrive at similar conclusions to the interpretation of results here.

I thought the conclusion that local inputs also reflect long range anatomical gradients is exciting and interesting; felt that perhaps it could have been better highlighted. The below quote for example packs lots of information into a few clauses with relatively little discussion.

"This thus suggests that local inputs to cluster 1 (SNr, SNl, mRT, SPTg, SubB, PnC, RI, PT, InC) principally target VTADA→NAcLat cells, those in cluster 2 (PSth, IC, RPC, VTA PBP, PN, IF, MnR, Rli, SNc, PMnR, VTg, MPL, MM, SuMx) predominately innervate VTADA→Amygdala neurons, and those in cluster 3 (IPN, SuM, SC, PnO, ATg, RMg, PPTg, RRF, RtTg, MiTg, DLL) are biased onto VTADA→NAcMed or VTADA→mPFC cells. "

Supp Figure 1 variance is so high and under sampled it is difficult to be confident in conclusions same with correlation in Figure 1. I think it is ok to report, but perhaps some discussion of limitations of this quite small dataset are warranted as a caveat.

---

## [Author Response]

Essential revisions:(1) For maximal impact and for adherence to eLife reporting standards it will be essential to release as much of these data as possible into the public domain, including at the minimum detailed tables with raw data quantification, but also consider sharing analysis code (e.g., for UMAP), as well as image files.

I am happy to deposit my data for public use. I plan to use Zenodo to do so (10.5281/zenodo.6407281).

(2) Please clarify important points raised below. No new experiments are necessary – but please provide thoughtful responses to the following requests and issues.2.1 Provide more background/references to explain the problem of low background levels of TVA expression. In theory, there should be zero Cre-independent expression of a functional TVA. Can the author please provide some explanation, or clarification, as to why this Cre-independent TVA expression (even if only a miniscule quantity) actually happens and is unavoidable? The paper is really all about the usage of the TC66T TVA variant, so this section should educate readers on the relevant nuances.Also please include references to other strategies to address this problem. Previously published studies have attempted to address the problem of low background levels of TVA expression and non-Cre dependent labeling of inputs near injection sites. One solution to prevent this ectopic expression of TVA is to titrate concentration of helper virus for efficient labeling of starter cells while minimizing non-Cre mediated expression, e.g. Wickersham's group https://doi.org/10.3389/fnsyn.2020.00006. Overall, please include a discussion of alternative, potentially simpler solutions. Please address how volume, titer, AAV serotype, promoter, and other batch effects could alter relative to AAV helper virus expression change the background level of non-Cre dependent TVA infection. Without varying these parameters extensively, especially given the diverse applications of this technology in hundreds of neuroscience labs, it is challenging to evaluate the depth of the problem with mapping local inputs.

I would like to thank the reviewers for their thoughtful comments. I am not the first to report the issue of off-target expression of FLEx/DIO AAV vectors, however I appreciate that this issue is not yet sufficiently appreciated by the community. I have now included an explanation of alternative approaches for reducing off-target rabies infection in the Discussion section, as well as general considerations about variables that could affect viral infectivity/expression. This can be found in the discussion, lines 605-690.

2.2 Please clearly define what is and what is not meant by the word 'input'Throughout the paper the author reports % of inputs as a quantity. But an 'input' can mean many things and more clarity is required to reduce confusion. Ideally, a report of % of inputs to a VTA DA neuron would relate to the actual fraction of inputs. For example if a given VTA neuron had 100 synapses, what fraction comes from SNR or STN or VTAgaba or VTAda? This is a per-neuron value, but even across a population of neurons an equivalent analysis could hold – if 100 neurons collectively receive 10,000 inputs, what fraction of those synapses come from a given location. This notion of input is not what is being reported here. Instead, if I understand correctly, it's the number of retrogradely labeled neurons from a given 'area' divided by the total number of retrogradely labeled neurons. It's unclear what physiological relevance this value has, for several reasons. First, a given neuron could project to anywhere between 1-100% of the starter cells. In the present analysis it's an input either way. Second, a given single neuronal input could be weak or strong, distal dendrite or somatic. Given the methods of this paper, synaptic strength is irrelevant. The author is likely fully aware of these issues – which may seem obvious – but I think they should be spelled out in the introduction when the word 'input' is defined, and again in the discussion as caveats for interpretation.

Thank you for raising this issue – I fully agree and indeed have raised these issues several times in previous manuscripts. However I agree that since this study so heavily relies on one-step RABV tracing that it is worthwhile to revisit here.

I added a line in the introduction (lines 79-80) to define what I measured, and then added a paragraph in the discussion (lines 752-779) about what one can and cannot infer from RABV one-step mapping experiments, including the issues raised above by the reviewer.

Third, the fraction of inputs in this paper is labeled as per anatomical region, established in the literature, but possibly meaningless for function. For example, eyeballing the figures from this paper or even the past work show a continuous smear of VTA-projecting neurons that follow various mediolateral, dorsoventral, anteroposterior gradients. The 'boundaries' between areas may or may not reflect anything meaningful beyond the history of neuroscientists drawing lines. Single cell sequencing and better pathway specific demarcation will reinforce the validity of these lines or reveal them to be irrelevant. Again, the only request for future versions of this paper is that caveats be made about how seriously to accept these boundaries. For example, does a VTA DA projecting neuron in the STN that is part of a continuous smear of neurons that reaches into the LH, 'know' that it is in the STN – especially if the inputs to these structures are also overlapping? Ideally the author could contextualize the findings from this work into some principle of organization of neuronal projections into VTA – beyond a simple well established adherence to a weak topography that respects mediolateral gradients.2.3 For this paper – this last issue also relates to confusion over the subterritories of VTA, where it is claimed that starter cells were found in "PBP nucleus of the VTA, 15 {plus minus} 1% in the VTA PN, 11 {plus minus} 2% in the SNc, 6 {plus minus} 1% in the VTA IF, 323 3 {plus minus} 1% in the Rli, and 2 {plus minus} 0.3% in all other regions." Yet the figure shows a continuous smear of neurons without clear nuclei or boundaries. While it is laudable that the author is attempting to achieve rigor of identification within VTA sub territories, more citations or background is necessary to justify them. And how do these acronyms relate to the catecholamine A1-A13 nomenclature? Are these boundaries just lines drawn by past researchers? Are they supported by clearcut distinctions in projection patterns, by distinct genetic identity as revealed by single cell sequencing? More clarification and background is necessary for VTA subdivisions, ideally with how they relate (or don't) to distinct targets. Finally, even when this issue of VTA sub-territories is addressed, the delineation of VTA into these subterritories is out of synch with the later portion of the paper that motivates mediolateral distinctions in input/output. Can these sections be better unified?

I would like to thank the reviewers for their thoughtful insights. Indeed, I believe that there are spatial patterns in the brain that sometimes cross and supersede anatomical boundaries defined by neuroanatomists. However, using these boundaries as spatial guideposts helps in describing and detailing our results. As for the DA neurons in the VTA, these regions are all part of A10. In order to provide a rationale for the existence of VTA subregions, I have added several lines of text in the results (lines 153-159) that highlight features of these subregions of the VTA, including several classical and more modern citations detailing different aspects of these subdivisions. I agree that they are somewhat arbitrary, but are based on cytoarchitectural features and connectivity patterns that differ between these subnuclei.

I have also added a subsection in the discussion to address the importance of boundaries within the VTA and input regions, with the subheader Anatomical definition of VTA subregions and input sites (lines 692-728). The focus of this text is to highlight, as the reviewer indicates, that many of the anatomical boundaries that we use are based on classical definitions and limited by what neuroanatomists had to use at the time. While it is worthwhile and indeed necessary to refer to these definitions in our work in order to compare our work to previous literature, it is also worth revisiting these boundaries in light of new emerging information (connectivity, molecular, etc.).

2.4 In Figure 2, the author's use of dimensionality reduction to analyze the relationship across local and long-range inputs is compelling and offers new ways of dissecting large-scale monosynaptic rabies tracing data. The author used z-score of fractional counts for UMAP input data and this ultimately aligns to spatial location of input fibers within the VTA. Considering that the UMAP input data is fractional counts for the number of input cells, does the corresponding spatial location of fibers relate to the density of fibers or fibers that are contacting specific subsets of VTA neurons?

The spatial location of fibers relates to the density of fibers. Our hypothesis is that the extent of rabies labeling is largely related to the density of fibers (Beier et al., Cell Reports 2019; Derdeyn et al., 2022). That may not be surprising, and on one hand suggests that RABV may spread promiscuously to nearby fibers. However I provided evidence that this is likely not the case (Beier et al., Cell Reports 2019). I thus find it likely that higher density of fibers in different locations in the VTA would correspond to elevated numbers of inputs onto VTA cell subtypes located in those regions. However, I did not test this directly, and other methods such as slice electrophysiology combined with optogenetics would be required to answer this question.

2.5. The conclusion that local inputs also reflect long range anatomical gradients is exciting and interesting, but it can be better highlighted. The below quote for example packs lots of information into a few clauses with relatively little discussion."This thus suggests that local inputs to cluster 1 (SNr, SNl, mRT, SPTg, SubB, PnC, RI, PT, InC) principally target VTADA→NAcLat cells, those in cluster 2 (PSth, IC, RPC, VTA PBP, PN, IF, MnR, Rli, SNc, PMnR, VTg, MPL, MM, SuMx) predominately innervate VTADA→Amygdala neurons, and those in cluster 3 (IPN, SuM, SC, PnO, ATg, RMg, PPTg, RRF, RtTg, MiTg, DLL) are biased onto VTADA→NAcMed or VTADA→mPFC cells. "Please expand this section, perhaps in the discussion, to include potential functional significance and/or discovery of developmental principles consistent with these results.

I expanded this section (lines 525-539) to discuss some of the implications of these results. This includes the advantage of using the UMAP-based approach to identify connectivity relationships that may otherwise be difficult to define using other retrograde or anterograde methods, the overall suggestion of a medial-lateral gradient (though not clearly so for all brain regions, as noted), and the potential functional association of regions with similar functions. I also allude to issues in neuroanatomical mapping, as what one can say functionally about many of these brain regions, especially the smaller ones, is limited because we simply do not know that much about what they do at this time.

Please ensure your manuscript complies with the eLife policies for statistical reporting: https://reviewer.elifesciences.org/author-guide/full "Report exact p-values wherever possible alongside the summary statistics and 95% confidence intervals. These should be reported for all key questions and not only when the p-value is less than 0.05."

I only performed one statistical analysis, in Figure 1H. The p-value was greater than 0.05. The n has been added.

If you have not already done so, please include a key resource table.

The table has now been added.

Reviewer #2 (Recommendations for the authors):Other:– Several figures have the smallest font of illegible size, even with zoomed in review.

I scaled up each figure 120%, except for Figure 2 where this would not have been possible given it was already at ~maximum width. Therefore I split Figure 2 into two separate figures and scaled up both independently. Providing high-resolution.tif files will also help to make the text in the figures legible, which I did.

– Formalize "local" and "long distance" in microns from the injection site.

This has been clarified. See lines 827-833.

– State the volume of rabies virus injected in the method section.

500 nL. This has been included in the methods; see line 813.

– Many previous studies inject rabies virus three weeks after helper virus, but this study waits only 2 weeks. What is the justification for this?

I had found that 2 weeks was sufficient to enable robust RABV labeling of starter cells and spread to inputs. I have found that input labeling scales approximately linearly with the number of labeled input cells, and given as we already have many thousands of labeled inputs, we would derive no further benefit from extending this time period and labeling potentially more input neurons. I cited several publications from the same period (from Liqun Luo’s lab, where I did my postdoc and original RABV tracing work) that use the same incubation period. See lines 814-816.

– The legend for supplemental figure 3 is vague. It lacks scale bars and does not specify what the thick gray lines are, or difference between image on left vs. right.

I added clarification about the sources of the images and what each represented. See lines 312-314. A scale bar has also been added.

– Lines 350-351, Figure 1 does not explicitly show that 10% of total inputs to VTA DA neurons are from local GABAergic neurons.

I originally worded it that way because throughout most of the VTA, I could not detect inputs from DA cells onto other DA cells as most DA cells near the injection site expressed mCherry (hence I had to take the extremes of the structure for Figure 5). Therefore, the majority of remaining non-DA cells in the VTA are GABAergic (with some glutamatergic cells). Nonetheless I agree that this wording was unnecessarily confusing. I therefore changed it to the following:

Local inputs from the VTA comprised about 10% of the total inputs to VTA^DA^ cells (inputs from the VTA-IF, VTA-PBP, and VTA-IF, Figure 1); see lines 387-388.

– Figure 3C is missing a scale bar.

A scale bar has been added.

– Figure 4b-e, images are missing scale bar and the legend does not specify what the white arrows represent.

A scale bar has been added and the arrows clarified.

– Line 367, "nearly" should be "nearby"?

Thank you, this has been corrected.

Reviewer #3 (Recommendations for the authors):Comparisons to prior data with UMAP was tough to draw conclusions from (Supp Figure 2) as noted by the authors. However, the augmented analysis using recent work was nice and indeed built confidence in the conclusion that the datasets were likely comparable. A little clarity about how conclusions were drawn would be helpful. Of course additional subjects to increase the N would be helpful, but understandable that Bender chose not to do further samples.

I added several lines of text in the Results section about how these data were interpreted and the conclusions that I drew from them. See lines 184-188.

"To test if the local inputs we identified indeed projected laterally to the VTA, we mapped out the relative innervation by the SNr, mRT, and PAG of the VTA across the medial-lateral gradient, as done previously for long-range inputs, using the Allen Mouse Brain Connectivity Atlas (Figure 2E) (Beier et al., 2019). We indeed observed that three selected inputs in cluster 1 – the SNr, mRT, and PAG – displayed a lateral bias in the VTA (Figure 2F, Supplemental Figure 3). While extensive efforts have mapped the inputs, including inhibitory cell inputs to the VTA (Geisler et al., 2007; Geisler and Zahm, 2005; Phillipson, 1979; Sesack and Grace, 2010; Swanson, 2000; Zahm et al., 2011), these methods lacked the connectivity information afforded by RABV."I agree with this point and just use this as a jumping off point. There is a reasonable number of studies that have attempted to measure functional connectivity with electrophysiology/optogenetic or reconstruction of axons from local GABAergic neurons (e.g. from SNr) and yet those papers are little discussed or cited. Might be relevant given that they arrive at similar conclusions to the interpretation of results here.

Thank you for this suggestion. I am aware of a substantial amount of work on mapping inhibitory inputs to the VTA, including those selectively modified by particular neurotransmitters or drugs. However I chose to focus my study mostly on localizing the GABAergic inputs to VTA^DA^ cells, agnostic to the functional implications of our findings given the limitations of RABV mapping as a method that are now highlighted more extensively in the manuscript. That being said, if there are particular citations that the reviewer feels I should discuss, I am happy to do so.

I thought the conclusion that local inputs also reflect long range anatomical gradients is exciting and interesting; felt that perhaps it could have been better highlighted. The below quote for example packs lots of information into a few clauses with relatively little discussion."This thus suggests that local inputs to cluster 1 (SNr, SNl, mRT, SPTg, SubB, PnC, RI, PT, InC) principally target VTADA→NAcLat cells, those in cluster 2 (PSth, IC, RPC, VTA PBP, PN, IF, MnR, Rli, SNc, PMnR, VTg, MPL, MM, SuMx) predominately innervate VTADA→Amygdala neurons, and those in cluster 3 (IPN, SuM, SC, PnO, ATg, RMg, PPTg, RRF, RtTg, MiTg, DLL) are biased onto VTADA→NAcMed or VTADA→mPFC cells. "Supp Figure 1 variance is so high and under sampled it is difficult to be confident in conclusions same with correlation in Figure 1. I think it is ok to report, but perhaps some discussion of limitations of this quite small dataset are warranted as a caveat.

I have included a discussion of the caveats of this study, including the limited n, at the end of the Discussion section. See lines 730-737.